# LLaVA-Med: Training a Large Language-and-Vision Assistant for Biomedicine in One Day

**Chunyuan Li***, **Cliff Wong***, **Sheng Zhang***, **Naoto Usuyama**, **Haotian Liu**, **Jianwei Yang**
**Tristan Naumann**, **Hoifung Poon**, **Jianfeng Gao**
Microsoft
https://aka.ms/llava-med

## Abstract

Conversational generative AI has demonstrated remarkable promise for empowering biomedical practitioners, but current investigations focus on unimodal text. Multimodal conversational AI has seen rapid progress by leveraging billions of image-text pairs from the public web, but such general-domain vision-language models still lack sophistication in understanding and conversing about biomedical images. In this paper, we propose a cost-efficient approach for training a vision-language conversational assistant that can answer open-ended research questions of biomedical images. The key idea is to leverage a large-scale, broad-coverage biomedical figure-caption dataset extracted from PubMed Central, use GPT-4 to self-instruct open-ended instruction-following data from the captions, and then fine-tune a large general-domain vision-language model using a novel curriculum learning method. Specifically, the model first learns to align biomedical vocabulary using the figure-caption pairs as is, then learns to master open-ended conversational semantics using GPT-4 generated instruction-following data, broadly mimicking how a layperson gradually acquires biomedical knowledge. This enables us to train a **L**arge **L**anguage **a**nd **V**ision **A**ssistant for Bio**Med**icine (LLaVA-Med) in less than 15 hours (with eight A100s). LLaVA-Med exhibits excellent multimodal conversational capability and can follow open-ended instruction to assist with inquiries about a biomedical image. On three standard biomedical visual question answering datasets, fine-tuning LLaVA-Med outperforms previous supervised state-of-the-art on certain metrics. To facilitate biomedical multimodal research, we will release our instruction-following data and the LLaVA-Med model.

## 1 Introduction

Parallel image-text data is abundantly available in the general domain, such as web images and their associated captions. Generative pretraining has proven effective to leverage this parallel data for self-supervised vision-language modeling, as demonstrated by multimodal GPT-4 [35] and open-sourced efforts such as LLaVA [27]. By instruction-tuning models to align with human intents based on multimodal inputs, the resulting large multimodal models (LMMs) exhibit strong zero-shot task completion performance on a variety of user-oriented vision-language tasks such as image understanding and reasoning, paving the way to develop general-purpose multimodal conversational assistants [22, 2, 23, 10].

While successful in the general domains, such LMMs are less effective for biomedical scenarios because biomedical image-text pairs are drastically different from general web content. As a result, general-domain visual assistants may behave like a layperson, who would refrain from answering

*Equal Contribution

37th Conference on Neural Information Processing Systems (NeurIPS 2023) Track on Datasets and Benchmarks.

biomedical questions, or worse, produce incorrect responses or complete hallucinations. Much progress has been made in biomedical visual question answering (VQA), but prior methods typically formulate the problem as classification (*e.g.,* among distinct answers observed in the training set) and are not well equipped for open-ended instruction-following. Consequently, although conversational generative AI has demonstrated great potential for biomedical applications [20, 33, 19], current investigations are often limited to unimodal text.

In this paper, we present **L**arge **L**anguage **a**nd **V**ision **A**ssistant for Bio**Med**icine (LLaVA-Med), a first attempt to extend multimodal instruction-tuning to the biomedical domain for end-to-end training of a biomedical multimodal conversational assistant. Domain-specific pretraining has been shown to be effective for biomedical natural language processing (NLP) applications [18, 15, 11, 31] and biomedical vision-language (VL) tasks [16, 7, 41, 51, 9]. Most recently, large-scale biomedical VL learning has been made possible by the creation of PMC-15M [51], a broad-coverage dataset with 15 million biomedical image-text pairs extracted from PubMed Central[1]. This dataset is two orders of magnitude larger than the next largest public dataset, MIMIC-CXR [16], and covers a diverse image types. Inspired by recent work in instruction-tuning [37, 27], LLaVA-Med uses GPT-4 to generate diverse biomedical multimodal instruction-following data using image-text pairs from PMC-15M, and fine-tune a large biomedical-domain VL model [27] using a novel curriculum learning method.

Specifically, our paper makes the following contributions:

- *Biomedical multimodal instruction-following data*. We present a novel data generation pipeline to create diverse (image, instruction, output) instances, by sampling biomedical image-text pairs from PMC-15M and using GPT-4 to create instructions from the text alone (which becomes the intended output). This requires zero manual annotations and creates an extremely diverse visual instruction-following dataset by piggybacking on PMC-15 that covers the full spectrum of research findings over biomedical images.
- *LLaVA-Med*. We propose a novel curriculum learning method for adapting LLaVA [27] to the biomedical domain using our self-generated biomedical multi-modal instruction-following dataset. Specifically, we first fine-tune LLaVA to align biomedical vocabulary using the image-text pairs as is (with the generic instruction that simply asks for a description of the image). We then continue training the model using our self-generated instruction-following data to learn open-ended conversational semantics. In this way, we were able to train LLaVA-Med in less than 15 hours with eight A100s. Our empirical study validates the effectiveness of domain-specific instruction-tuning, and reveals best practice and interesting findings for adapting multimodal conversational assistant to high-value verticals. On well-established biomedical VQA datasets, fine-tuning LLaVA-Med often outperforms supervised state-of-the-art (SoTA).
- *Open-source*. To facilitate research in biomedical multimodal learning, we will release the following assets to the public: the biomedical multimodal instruction-following dataset and the codebase for data generation and model training.

## 2 Related Work

**Biomedical Chatbots.** Inspired by ChatGPT [34]/GPT-4 [35] and the success of open-sourced instruction-tuned large language models (LLMs) in the general domain, several biomedical LLM chatbots have been developed, including ChatDoctor [49], Med-Alpaca [13], PMC-LLaMA [47], Clinical Camel [1], DoctorGLM [48], and Huatuo [46]. They are initialized with open-sourced LLM and fine-tuned on customized sets of biomedical instruction-following data. The resulting LLMs emerge with great potential to offer assistance in a variety of biomedical-related fields/settings, such as understanding patients' needs and providing informed advice.

To our knowledge, Visual Med-Alpaca [42] is the only existing multimodal biomedical chatbot that accepts image inputs. Though Visual Med-Alpaca and the proposed LLaVA-Med share a similar input-output data format, they differ in key aspects: $(i)$ *Model architectures.* LLaVA-Med is an end-to-end neural model and Visual Med-Alpaca is a system that connect multiple image captioning models with a LLM, using a classifier to determine if or which biomedical captioning model is responsible for the image. The text prompt subsequently merges the converted visual information with the textual query, enabling Med-Alpaca to generate an appropriate response. $(ii)$ *Biomedical*

[1] https://www.ncbi.nlm.nih.gov/pmc/

*instruction-following data.* While Visual Med-Alpaca is trained on 54K samples from limited biomedical subject domains, LLaVA-Med is trained a more diverse set.

**Biomedical Visual Question Answering.** An automated approach to building models that can answer questions based on biomedical images stands to support clinicians and patients. To describe existing biomedical VQA methods, we make a distinction between discriminative and generative methods. For discriminative methods, VQA is treated a classification problem: models make predictions from a predefined set of answers. While discriminative methods yield good performance, they deal with closed-set predictions [14], and require mitigation when a customized answer set is provided in at inference [24, 51, 9]. The discriminative formulation is suboptimal towards the goal of developing a general-purpose biomedical assistant that can answer open questions in the wild. To this end, generative methods have been developed to predict answers as a free-form text sequence [5, 30, 44]. Generative methods are more versatile because they naturally cast the close-set questions as as special case where candidate answers are in language instructions.

**Data-Centric Paradigms.** LLaVA-Med is similar to prefix tuning of language models (LMs) in [44] in that a new trainable module connects frozen image encoder and causal LM. In [44], a three-layer MLP network is used to map the visual features into a visual prefix, and the pre-trained LM are GPT2-XL [40], BioMedLM [45] and BioGPT [31], with size varying from 1.5B to 2.7B. By contrast, LLaVA-Med uses a linear projection and a 7B LM [8, 43]. Most importantly, we undertake *data-centric* paradigm, while all existing methods are *model-centric*. [44] focuses efforts on exploring various modeling choices. Our main contributions instead comprise proposing a novel data generation method that uses GPT-4 to self-instruct biomedical multimodal instruction-following data using freely-available broad-coverage biomedical image-text pairs extracted from PubMed Central [51].

## 3 Biomedical Visual Instruction-Following Data

There are a lack of multimodal biomedical datasets to train an instruction-following assistant. To fill this gap, we create the first dataset of its kind from widely existing biomedical image-text pairs, through a machine-human co-curation procedure. It consists of two sets, concept alignment and instruction-following, which are used at different training stages, described in Section 4.

**Biomedical Concept Alignment Data.** For a biomedical image $\mathbf{X}_{\texttt{v}}$ and its associated caption $\mathbf{X}_{\texttt{c}}$, we sample a question $\mathbf{X}_{\texttt{q}}$, which asks to describe the biomedical image. With $(\mathbf{X}_{\texttt{v}}, \mathbf{X}_{\texttt{c}}, \mathbf{X}_{\texttt{q}})$, we create a single-round instruction-following example:

$$\texttt{Human}: \mathbf{X}_{\texttt{q}}\ \mathbf{X}_{\texttt{v}}\texttt{<STOP>}\backslash\texttt{n}\ \texttt{Assistant}: \mathbf{X}_{\texttt{c}}\texttt{<STOP>}\backslash\texttt{n} \tag{1}$$

Depending on the length of caption, the question that is sampled either asks to describe the image *concisely* or *in detail*. Two lists of questions are provided in Appendix A. In practice, 25% of captions have length less than 30 words in PMC-15M [51], and thus 30 words is used as the cutoff point to determine which list to choose. We sample 600K image-text pairs from PMC-15M. Though this dataset only presents one-single task instructions, *i.e.,* image captioning, it contains a diverse and representative set of biomedical concept samples from the original PMC-15M [51].

**Biomedical Instruction-Tuning Data.** To align the model to follow a variety of instructions, we present and curate diverse instruction-following data with multi-round conversations about the provided biomedical images, by prompting language-only GPT-4. Specifically, given an image caption, we design instructions in a prompt that asks GPT-4 to generate multi-round questions and answers in a tone as if it could see the image (even though it only has access to the text). Sometimes the image caption is too short for GPT-4 to generate meaningful questions and answers. To provide more context regarding the image, we also create a prompt that includes not only captions but also sentences from the original PubMed paper that mentions the image. We also manually curate few-shot examples in the prompt to demonstrate how to generate high-quality conversations based on the provided caption and context. See Appendix A.2 for the prompt and few-shot examples. To collect image captions and their context, we filter PMC-15M to retain the images that only contain a single plot. From them, we sample 60K image-text pairs from the five most common imaging modalities: CXR (chest X-ray), CT (computed tomography), MRI (magnetic resonance imaging), histopathology, and gross (*i.e.,* macroscopic) pathology. We then extract sentences that mention the image from the

**Biomedical Visual Instruction-Following Example**

**Caption:**

Figure 4: Contrast-enhanced CT scan of the chest for patient #1. A large cavitating lesion was identified in the right upper lobe with pleural involvement and likely extension into the chest wall. In addition, right hilar lymphadenopathy was suspected. This was the primary malignant lesion in this patient.

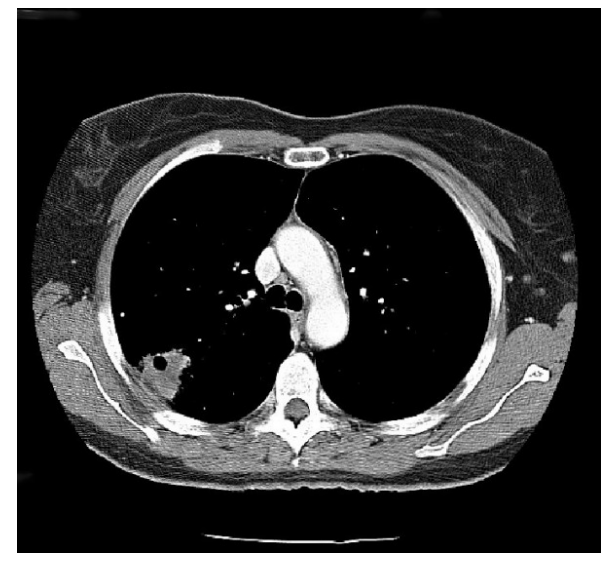

**Sentences with in-line figure mentions**:

- Computerized tomography ( CT ) scans of the chest ( Figure 4 ) were obtained.
- ( Figure 4 ) and demonstrated a large cavitating lesion in the posterior aspect of the right upper lobe with probable pleural involvement likely representing the primary lesion.

**Conversation:**

```
User: Where is the lesion located?

Assistant: The lesion is located in the posterior aspect of the right upper lobe of the lung.

User: Is it extending into the chest wall?

Assistant: Yes, the CT scan suggests that the large cavitating lesion in the right upper lobe is
likely extending into the chest wall.

User: Can you explain more about the cavitating lesion?

Assistant: Certainly. A cavitating lesion refers to an abnormal mass or growth within the lung
tissue that has a cavity or hollow space inside it. This cavity can be formed due to the necrosis
(death) of the tissue inside the lesion, leading to the breakdown and disintegration of the
affected tissue. Cavitating lesions can be caused by various factors, such as infections (like
tuberculosis or fungal infections), autoimmune diseases, or malignancies (cancer). In the context
of the CT scan provided, the large cavitating lesion in the right upper lobe of the lung is
suggestive of a malignant (cancerous) growth. The cavity within the lesion may be a result of the
rapid and uncontrolled growth of cancer cells, which can lead to the destruction of surrounding
healthy tissue. It's important to note that further diagnostic tests, such as a biopsy, would be
required to confirm the malignancy and determine the specific type of cancer involved.
```

Figure 1: An instance of our GPT-4 generated instruction-following data. Top: The figure and caption were extracted from a PubMed Central full-text article [38], along with the corresponding citances (mentions of the given figure in the article). Bottom: The instruction-following data generated by GPT-4 using the text only (caption and citances). Note that the image is not used to prompt GPT-4; we only show it here as a reference.

original PubMed paper as additional context to the caption, inspired by the observations that external knowledge helps generalization [21, 28].

An example of instruction-following data is shown in Figure 1 shows, and the data statistics is shown Figure 2. We have produced three versions of instruct data when iteratively improving the data quality: $(i)$ *60K-IM*. The aforemenioned dataset that considers inline mentions (IM) as the context. $(ii)$ *60K*. A dataset of similar size (60K samples) without IM in self-instruct generation. $(iii)$ *10K*. A smaller dataset (10 samples) without IM. They are used to ablate our data generation strategies and their impact on trained LLaVA-Med in experiments.

## 4 Adapting Multimodal Conversational Models to the Biomedical Domain

We employ LLaVA, a general-domain multimodal conversation model [27], as the initial general-domain LM, and continuously train the model to the biomedical domain. The same network architecture is utilized, where a linear projection layer connects the vision encoder and the language model. For LLaVA-Med model training, we use a two-stage procedure, illustrated in Figure 3.

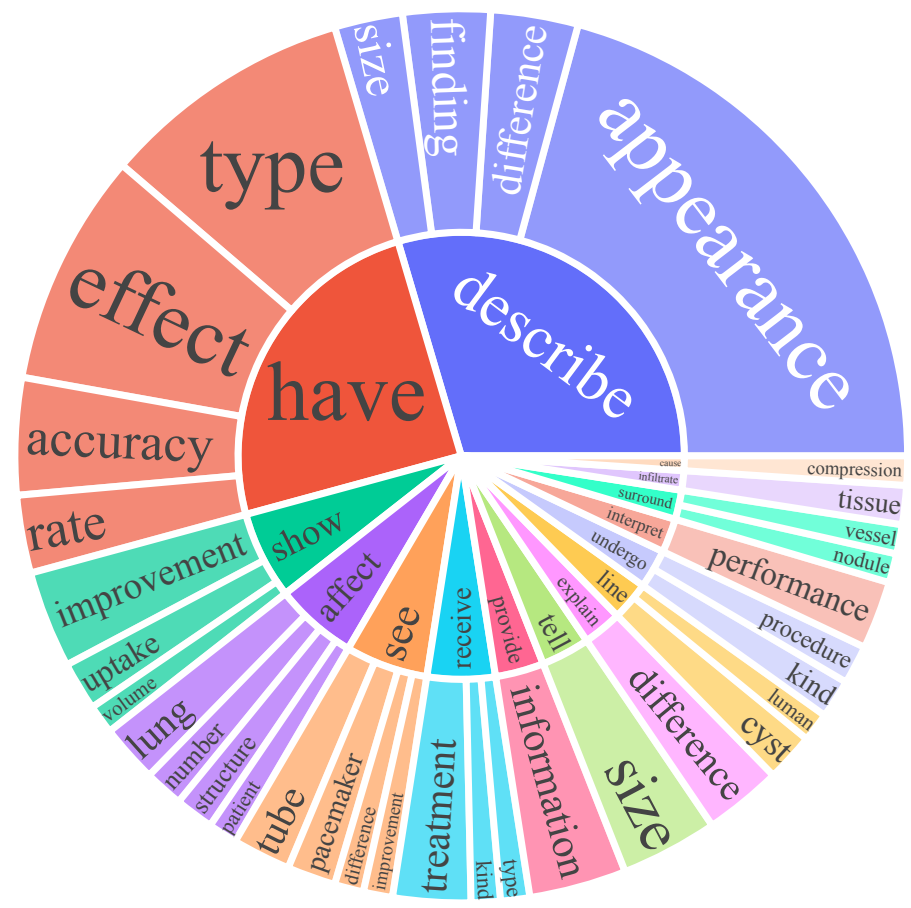

(a) Instruction

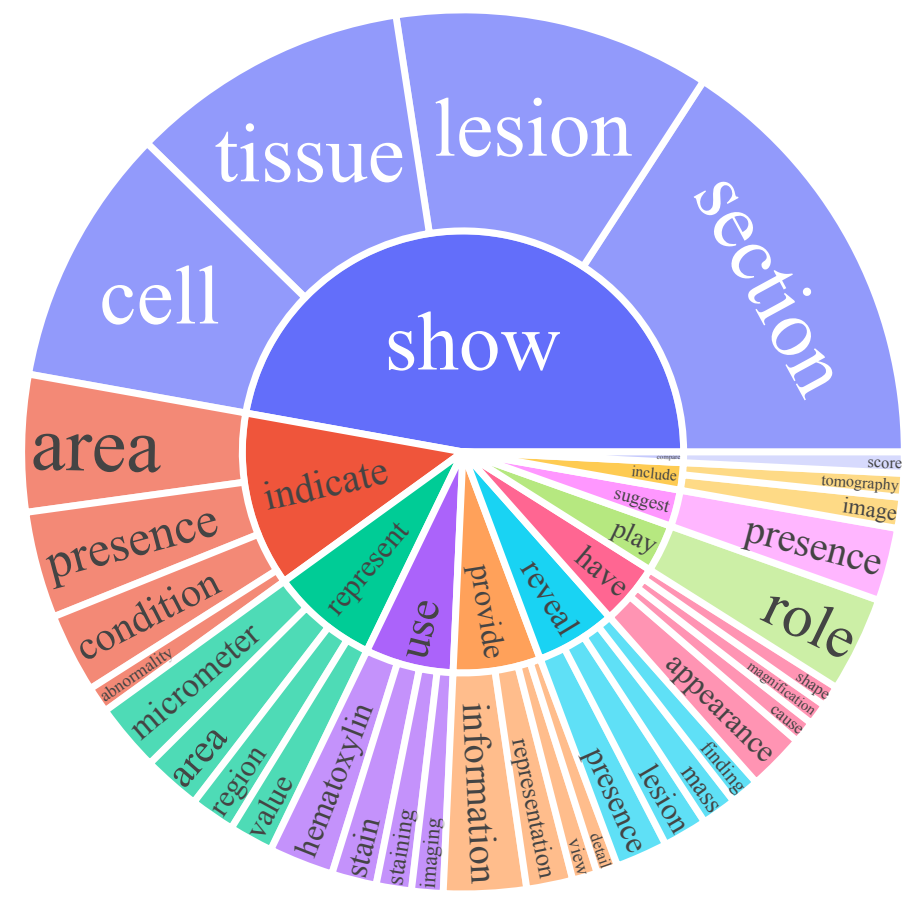

(b) Responses

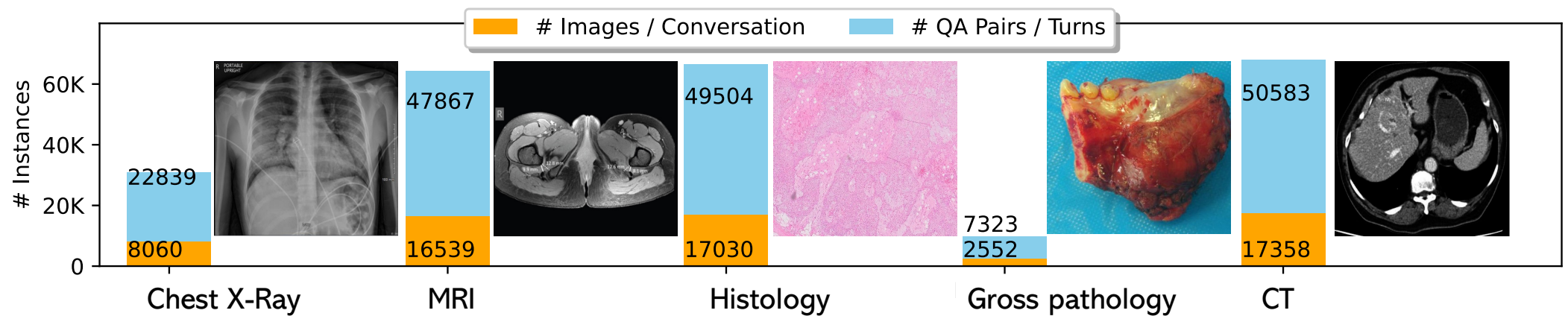

(c) Frequencies of images and QA pairs on the five domains.

Figure 2: The data statistics of biomedical multimodal instruction-following data: (a,b) The root verb-noun pairs of instruction and responses, where the inner circle of the plot represents the root verb of the output response, and the outer circle represents the direct nouns. (c) The distribution of images and QA pairs on the five domains, one image is shown per domain. The domain example images are from [3, 36, 4, 32, 50].

**Stage 1: Biomedical Concept Feature Alignment.** To balance between concept coverage and training efficiency, we filter PMC-15M to 600K pairs. These pairs are converted to instruction-following data using a naive expansion method: instructions simply presents the task of describing the image. For each sample, given the language instruction and image input, we ask the model to predict the original caption. In training, we keep both the visual encoder and LM weights frozen, and only update the projection matrix. In this way, the image features of vast novel biomedical visual concepts can be aligned to their textual word embeddings in the pre-trained LM. This stage can be understood as expanding the vocabulary of aligned image-text tokens to the biomedical domain. This stage is critical when training LLaVA-Med from customized vision encoder and LLM, where LLaVA initialization is not available. However, when training LLaVA-Med by initializing from LLaVA, we find that this stage can be either skipped to save compute cost, or consider the strategy of removing $\mathbf{X}_\mathrm{q}$ in (1) and tuning LLM for higher performance. Please see the discussion in Appendix C.2.

**Stage 2: End-to-End Instruction-Tuning.** We only keep the visual encoder weights frozen, and continue to update both the pre-trained weights of the projection layer and LM. To train the model to follow various instructions and complete tasks in a conversational manner, we develop a biomedical chatbot by fine-tuning our model on the biomedical language-image instruction-following data collected in Section 3. As demonstrated in the experiments to be described later, the LLaVA-Med model at this stage is able to not only be served as a biomedical visual assistant to interact with users, but also achieve good zero-shot task transfer performance when evaluated on well-established biomedical VQA datasets.

**Discussion.** We discuss four favorable properties/implications of LLaVA-Med: $(i)$ *Affordable development cost.* Instead of scaling up data/model for the best performance, we aim to provide affordable and reasonable solutions with low development cost: it takes 7 and 8 hours for stage 1 and 2 on 8 40G A100 GPUs, respectively (see Table 5 for detailed numbers). $(ii)$ *A recipe for many domains.*

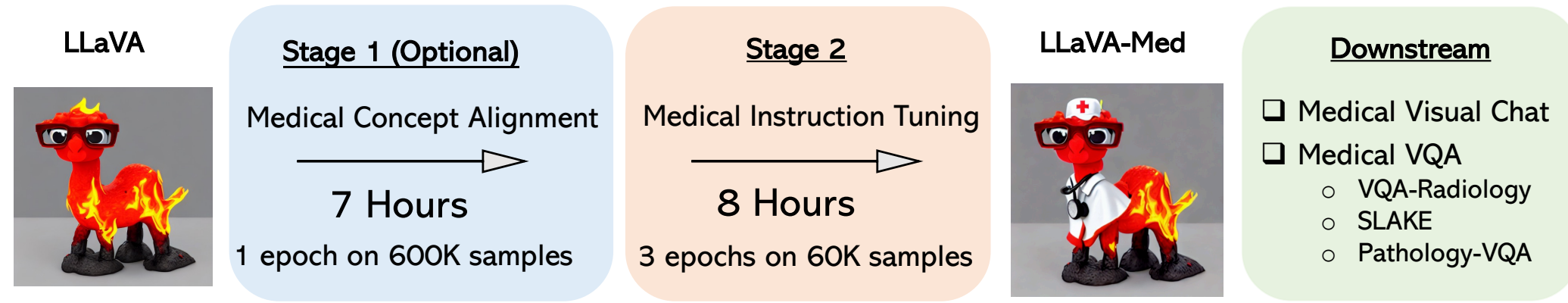

Figure 3: LLaVA-Med was initialized with the general-domain LLaVA and then continuously trained in a curriculum learning fashion (first biomedical concept alignment then full-blown instruction-tuning). We evaluated LLaVA-Med on standard visual conversation and question answering tasks.

Though this paper focuses on biomedical domains, the proposed adaptation procedure is generalizable to other vertical domains such as gaming and education, where novel concepts and domain knowledge are needed to build a helpful assistant. Similar to the *don't stop pre-training* argument in [12], we consider a scalable pipeline to create domain-specific instruct data from large unlabelled data, and advocate *don't stop instruction-tuning* to build customized LMM. $(iii)$ *Low serving cost.* While the model size of general LMM can be giant and serving cost can be prohibitively high, customized LMM has its unique advantages in low serving cost. $(iv)$ *Smooth Model Adaptation.* Alternatively, the network architecture allows us to initialize the vision encoder from BioMedCLIP [51], or initialize the language model from Vicuna [8], which may lead to higher performance. However, adapting from LLaVA smooth adaptation as a chatbot, where model's behaviors transit from layperson to a professional assistant that is able to provide helpful domain-specific response.

## 5 Experiments

We conduct experiments to study two key components, the quality of the produced multimodal biomedical instruction-following data, and performance of LLaVA-Med. We consider two research evaluation settings: (1) What is the performance of LLaVA-Med as an open-ended biomedcal visual chatbot? (2) How does LLaVA-Med compare to existing methods on standard benchmarks? To clarify, throughout the entire experiments, we only utilize the language-only GPT-4.

### 5.1 Biomedical Visual Chatbot

To evaluate the performance of LLaVA-Med on biomedical multimodal conversation, we construct an evaluation dataset with 193 novel questions. For this test dataset, we randomly selected 50 *unseen* image and caption pairs from PMC-15M, and generate two types of questions: conversation and detailed description. The conversation data is collected using the same self-instruct data generation pipeline as for the 2nd stage. Detailed description questions were randomly selected from a fixed set [27] of questions to elicit detailed description responses.

We leverage GPT-4 to quantify the correctness of the model answer to a question when given the image context and caption. GPT-4 makes a reference prediction, setting the upper bound answer for the teacher model. We then generate response to the same question from another LMM. Given responses from the two assistants (the candidate LMM and GPT-4), the question, figure caption, and figure context, we ask GPT-4 to score the helpfulness, relevance, accuracy, and level of details of the responses from the two assistants, and give an overall score on a scale of 1 to 10, where a higher score indicates better overall performance. GPT-4 is also asked to provide a comprehensive explanation the evaluation, for us to better understand the models. We then compute the relative score using GPT-4 reference score for normalization.

The results are reported in Table 1. LLaVA-Med with Stage-1 training alone is insufficient as a chatbot, as it loses its ability to follow diverse instructions, though biomedical concept coverage is improved. LLaVA-Med with the full two-stage training consistently outperforms the general domain LLaVA, and training with larger instruct data (from 10K to 60K samples) leads to higher performance. When inline mentions are considered in self-instruct, the generated data 60K-IM slightly improves the chat ability. The results demonstrate the effectiveness of the strategies in biomedical instruction-following data collection as well as the value of dataset assets. Overall, for the best LLaVA-Med, it matches the 50.2% performance of GPT-4. Note that GPT-4 generates response by considering

| | Question Types | | Domains | | | | | Overall |
|---|---|---|---|---|---|---|---|---|
| | Conversation | Description | CXR | MRI | Histology | Gross | CT | |
| (Question Count) | (143) | (50) | (37) | (38) | (44) | (34) | (40) | (193) |
| LLaVA | 39.4 | 26.2 | 41.6 | 33.4 | 38.4 | 32.9 | 33.4 | 36.1 |
| LLaVA-Med | | | | | | | | |
| Stage 1 | 22.6 | 25.2 | 25.8 | 19.0 | 24.8 | 24.7 | 22.2 | 23.3 |
| 10K | 42.4 | 32.5 | 46.1 | 36.7 | 43.5 | 34.7 | 37.5 | 39.9 |
| 60K | 53.7 | 36.9 | 57.3 | 39.8 | 49.8 | 47.4 | 52.4 | 49.4 |
| 60K-IM | 55.1 | 36.4 | 56.2 | 40.4 | 52.7 | 51.8 | 50.1 | 50.2 |

Table 1: Performance comparison of mulitmodal chat instruction-following abilities, measured by the relative score via language GPT-4 evaluation.

**Visual input example, Biomedical image from [6]**

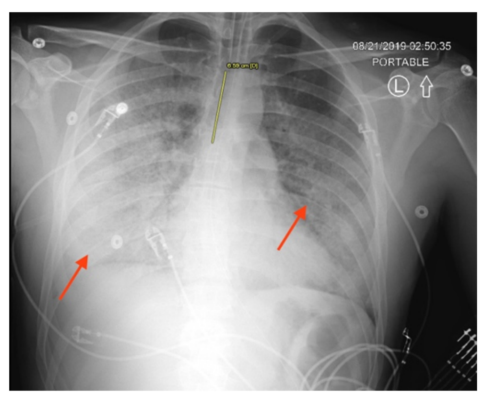

Caption: Chest x-ray (CXR) on Day 2 of admission post-intubation (yellow line showing the level of the endotracheal tube). Rapidly worsening ground-glass opacities are seen throughout the lungs with relative subpleural sparing (red arrows)

Source: https://www.ncbi.nlm.nih.gov/pmc/articles/PMC7398722/

| | |
|---|---|
| User | What is shown in this image? |
| LLaVA-Med | The image is a chest X-ray (CXR) that shows bilateral patchy infiltrates, which are areas of increased opacity in the lungs. These infiltrates can be indicative of various lung conditions, such as infections, inflammation, or other lung diseases. |
| User | Are there any devices implanted in this patient? |
| LLaVA-Med | Yes, it appears that the patient has several devices implanted. These devices may include endotracheal tubes, central venous catheters, and/or pulmonary artery catheters. These devices are typically used for monitoring and managing various medical conditions, as well as for providing support during surgical procedures. |
| User | What is shown in this image? |
| LLaVA | The image shows a ribcage with wires coming out of it, indicating that it is likely a medical illustration or diagram. |
| User | Are there any devices implanted in this patient? |
| LLaVA | Yes, there are several devices protruding from the patient's chest, which are likely medical devices used for treatment or monitoring. |
| User | What is shown in this image? |
| Language GPT-4 | The image shows a chest x-ray taken on Day 2 of a patient's admission after intubation. You can see the endotracheal tube marked by a yellow line, and rapidly worsening ground-glass opacities throughout the lungs. There is also relative subpleural sparing, which is indicated by red arrows. |
| User | Are there any devices implanted in this patient? |
| Language GPT-4 | Yes, there is an endotracheal tube implanted in the patient, as indicated by the yellow line in the image. |

Table 2: Example comparison of medical visual chat and reasoning capabilities. The language-only GPT-4 is considered as the performance upper bound, as the golden captions and inline mentions are fed into GPT-4 as the context, without requiring the model to understand the raw image.

ground-truth caption and golden inline mentions, without understanding the images. Though not a fair comparison between LMMs and GPT-4, GPT-4 is a consistent and reliable evaluation tool.

Recent studies [25, 29, 52] raises the self-enhancement bias issue of LLM evaluation, *i.e.,* favoring its own generations. While we agree with the existence of the bias, we believe that GPT-4 is a meaningful and consistent measurement in our settings. We always compare the answers from the candidate model against the GPT-4's answers when computing the relative socre, and the ranking of the resulting numbers is consistent, though the numbers themselves might be biased to GPT's answers. The conclusions of our ablation study can be obtained by considering the ranking. Further, by taking the self-enhancement bias into consideration for fairness, we expect that LLaVA-Med actually performs even closer to GPT-4 than the current numbers indicate.

In Table 2, we provide examples on the biomed visual conversations of different chatbots. LLaVA-Med precisely answers the questions with biomedical knowledge, while LLaVA behaves like a

| | VQA-RAD | | | SLAKE | | | PathVQA | | |
|---|---|---|---|---|---|---|---|---|---|
| Method | Ref | Open | Closed | Ref | Open | Closed | Ref | Open | Closed |
| *Supervised finet-tuning results with our own experiment runs* | | | | | | | | | |
| LLaVA | | 50.00 | 65.07 | | 78.18 | 63.22 | | 7.74 | 63.20 |
| LLaVA-Med (From LLaVA) | | 61.52 | **84.19** | | 83.08 | 85.34 | | 37.95 | **91.21** |
| LLaVA-Med (From Vicuna) | | 64.39 | 81.98 | | **84.71** | 83.17 | | 38.87 | **91.65** |
| LLaVA-Med (BioMed CLIP) | | 64.75 | 83.09 | | **87.11** | 86.78 | | 39.60 | **91.09** |
| *Representative & SoTA methods with numbers reported in the literature* | | | | | | | | | |
| VL Encoder–Decoder [5] | 71.49 | | 82.47 | | | | 71.49 | | 85.61 |
| Q2ATransformer [30] | 79.19 | | 81.20 | | | | 54.85 | | 88.85 |
| Prefix T. Medical LM [44] | | | | 84.30 | | 82.01 | 40.00 | | 87.00 |
| PubMedCLIP [9] | 60.10 | | 80.00 | 78.40 | | 82.50 | | | |
| BiomedCLIP [51] | 67.60 | | 79.80 | 82.05 | | 89.70 | | | |
| M2I2 [24] | 66.50 | | 83.50 | 74.70 | | 91.10 | 36.30 | | 88.00 |

Table 3: Comparison with prior state-of-the-art supervised methods. For open-set questions, we report the recall for our free-form text generation method in column *Open*. For closed-set questions, we report the accuracy in column *Closed*. Bold indicates LLaVA-Med achieves new SoTA. For open-ended questions, prior methods still formulate the problem as classification among distinct answers in the training set, which may overestimate their generalizability as these datasets are unusual in that the test answers are almost always present in training..

layperson, who hallucinate based on commonsense. Since the multimodal GPT-4 is not publicly available, we resort to language-only GPT-4 for comparison. We feed golden captions and inline mentions into GPT-4 as the context, it generates knowledgeable response through re-organizing the information in the conversational manner.

### 5.2 Performance on Established Benchmarks

**Datasets and Evaluation Metrics.** We train and evaluate LLaVA-Med on three biomedical VQA datasets. The data statistics are summarized in Table 8 in Appendix. For the closed-set questions, we report the accuracy/percentage of the ground-truth tokens that appear in the generated sequences. For open-set questions, we use recall to evaluate the ratio that ground-truth tokens appear in the generated sequences. In the literature, the open-set problem is formulated as an closed-set setting, where the unique training answers are considered as the answer candidates, from which the models can select to predict answers for testing questions. Since we do not provide any constraint for the responses to open-set questions, our formulation is closer to open-set nature, but is intrinsically harder.

**Comparisons with SoTA.** We compare LLaVA-Med with the general domain LLaVA and existing representative methods in Table 3. First, All LLaVA-Med variants outperform LLaVA. While the difference of language model initialization from LLaVA or Vicuna is minor, the initialization of vision encoder from BioMed CLIP is slightly better than from general-domain CLIP. Second, the fine-tuning performance of LLaVA-Med is higher than supervised SoTA on the closed-set questions on VQA-RAD and PathVQA. This validates LLaVA-Med's strong ability in following instruction to complete biomedical tasks, when clear instructions are provided (*e.g.,* , yes or no). Third, for open-set questions, LLaVA-Med achieves SoTA on SLAKE, while its performance is limited on other datasets, especially compared with existing methods. This is perhaps because the open-set biomedical questions can be ambiguous without constraining their excepted answer options. Meanwhile, evaluation of free-form text prediction for medical VQA remains as an open research problem that we are actively exploring.

**Ablation Studies.** To study the impact of our curated instruction data and hyper-parameters in the training pipeline, we report the performance of different model variants in Table 4 (a). Several findings are confirmed: $(i)$ LLaVA-Med consistently outperforms LLaVA by a large margin, indicating the effectiveness of our biomedical domain-specific adaptation. The performance gaps on zero-shot are larger than that in fine-tuned settings, showing that LLaVA-Med is clearly a better option than LLaVA when deploying one model for various scenarios in the wild. $(ii)$ Training longer in Stage 1 improves zero-shot transfer, but Stage 1 alone is not sufficient, because the single image captioning

| LLaVA-Med Model Variants | | | | VQA-RAD | | SLAKE | | PathVQA | | Average |
|---|---|---|---|---|---|---|---|---|---|---|
| Instruct | Stage 1 | Stage 2 | FT | Open | Closed | Open | Closed | Open | Closed | |
| *CLIP Vision Encoder [39], 7B Language Model* | | | | | | | | | | |
| 0 | 1 | 0 | 0 | 15.27 | 12.50 | 18.55 | 13.46 | 6.26 | 13.51 | 13.26 |
| 0 | 3 | 0 | 0 | 15.33 | 15.44 | 23.61 | 15.38 | 6.35 | 14.74 | 15.14 |
| 10K | 1 | 3 | 0 | 25.79 | 57.35 | 31.50 | 51.68 | 8.49 | 59.66 | 39.08 |
| 10K | 3 | 3 | 0 | 28.44 | 59.56 | 22.63 | 43.99 | 5.40 | 52.67 | 35.45 |
| 10K | 1 | 3 | 1 | 36.39 | 55.88 | 71.64 | 56.49 | 25.50 | 82.87 | 54.79 |
| 10K | 1 | 3 | 3 | 18.59 | 55.51 | 78.60 | 63.46 | 34.02 | 86.94 | 56.19 |
| 60K | 1 | 1 | 0 | 29.80 | 55.15 | 38.08 | 50.00 | 11.70 | 59.66 | 40.73 |
| 60K | 1 | 3 | 0 | 29.67 | 60.29 | 35.53 | 53.85 | 11.76 | 53.20 | 40.72 |
| 60K | 1 | 3 | 1 | 22.63 | 58.09 | 72.75 | 54.33 | 24.19 | 71.60 | 50.60 |
| 60K | 1 | 3 | 3 | 54.12 | 64.71 | 79.33 | 64.90 | 17.18 | 71.37 | 58.60 |
| 60K-IM | 1 | 1 | 0 | 29.67 | 61.40 | 38.44 | 52.40 | 11.41 | 56.24 | 41.59 |
| 60K-IM | 1 | 3 | 0 | 28.23 | 61.40 | 39.17 | 52.16 | 12.30 | 54.05 | 41.22 |
| 60K-IM | 1 | 3 | 1 | 28.61 | 56.25 | 70.58 | 54.57 | 11.17 | 59.19 | 46.73 |
| 60K-IM | 1 | 3 | 3 | 55.50 | 66.54 | 80.57 | 64.18 | 35.88 | 89.15 | 65.30 |
| 60K-IM | 1 | 3 | 9 | 66.26 | 80.88 | 82.30 | 84.86 | 37.59 | 91.54 | 73.90 |
| 60K-IM | 1 | 3 | 15 | 61.53 | 84.19 | 83.08 | 85.34 | 37.95 | 91.21 | 73.88 |
| 60K-IM | 1 | 3 | 18 | 61.37 | 81.25 | 84.24 | 83.17 | 37.88 | 91.39 | 73.22 |
| *CLIP Vision Encoder [39], 13B Language Model* | | | | | | | | | | |
| 60K-IM | 1 | 3 | 0 | 31.66 | 61.40 | 37.71 | 49.76 | 11.34 | 49.63 | 40.25 |
| 60K-IM | 1 | 3 | 9 | 64.58 | 77.94 | 84.97 | 85.58 | 38.82 | 92.39 | 74.05 |
| *BioMed CLIP Vision Encoder [51], 7B Language Model* | | | | | | | | | | |
| 60K-IM | 1 | 3 | 0 | 37.84 | 60.66 | 39.73 | 54.33 | 11.65 | 49.07 | 42.21 |
| 60K-IM | 1 | 3 | 9 | 64.75 | 83.09 | 87.11 | 86.78 | 39.60 | 91.09 | 75.40 |
| LLaVA | 0 | 0 | 0 | 20.74 | 59.19 | 26.82 | 50.24 | 8.74 | 45.65 | 35.23 |
| LLaVA | 0 | 0 | 3 | 50.00 | 65.07 | 78.18 | 63.22 | 7.74 | 63.20 | 54.57 |

(a) Ablation studies with varying number of training epochs at different stages. 60K-IM indicates the instruct data generated with inline mentions. The gray rows are zero-shot performance of LLaVA-Med trained with different instruct data, they are selected to show in Table 3.

| LLaVA-Med Model Variants | | | | | VQA-RAD | | SLAKE | | PathVQA | | Average |
|---|---|---|---|---|---|---|---|---|---|---|---|
| Description | Module | Stage 1 | Stage 2 | FT | Open | Closed | Open | Closed | Open | Closed | |
| *Training* **without** *Stage-1* | | | | | | | | | | | |
| N/A | N/A | 0 | 3 | 0 | 23.28 | 62.13 | 36.50 | 56.01 | 7.73 | 58.12 | 40.63 |
| *Training* **with** *Stage-1* | | | | | | | | | | | |
| Yes | Projection | 1 | 3 | 0 | 25.79 | 57.35 | 31.50 | 51.68 | 8.49 | 59.66 | 39.08 |
| Yes | LLM | 1 | 3 | 0 | 26.23 | 52.33 | 37.62 | 52.48 | 9.38 | 58.88 | 39.49 |
| No | LLM | 1 | 3 | 0 | 26.88 | 56.16 | 35.23 | 55.78 | 9.39 | 63.27 | 40.97 |

(b) The impact of Stage-1 training. All jobs are initialized with LLaVA. "Description" indicates whether the descrption text is included in input in Stage-1. "Module" indicates the trainable module in Stage-1. The numbers in green cells shows the effectiveness of Stage-1 training, though the average scores are similar.

Table 4: Quantitative results on three established biomedical VQA datasets. "Instruct" is the instruct dataset, "Stage 1" and "Stage 2" provides the number of training epochs for each stage, "FT" is Fine-Tuning.

| Stage 1 | | Stage 2 | | | VQA-RAD | | SLAKE | | PathVQA | |
|---|---|---|---|---|---|---|---|---|---|---|
| 1 | 3 | Instruct | 1 | 3 | 1 | 3 | 1 | 3 | 1 | 3 |
| 6.8 | 19.4 | 10K | 0.6 | 1.8 | 0.3 | 0.6 | 0.6 | 1.0 | 1.0 | 2.5 |
| | | 60K | 2.6 | 8.0 | | | | | | |

Table 5: Running time (hours) for 1 and 3-epoch training, with batch size 128 on eight A100 GPUs.

instruction in Stage 1 may encourage the model to lose its ability in follow diverse instructions. $(iii)$ Instruction-following data in Stage 2 is critical, and the performance is generally improved, when the instruct data amount increases from 10K to 60K. The 60K-IM data provides the best averaged zero-shot and fine-tuned performance, respectively, validating the effectiveness of considering inline mention as external knowledge in data creation. $(iv)$ Fine-tuning longer on downstream datasets till 9 epochs benefits the performance, especially on checkpoints with 3-epoch training in Stage 2. Increasing language model size from 7B to 13B improves the overall zero-shot performance and fine-tuned performance. We suggest practitioners to choose the appropriate quality-cost trade-off, by referring to the running time in Table 5. $(v)$ When downstream samples are available, fine-tuning itself provides the largest performance gain (54.57-35.23=19.34). However, by training with high quality instruct data such as 60K-IM in the Stage-2, we can further boost performance significantly (65.30-54.57=10.73). Stage-2 itself is not as effective as direct fine-tuning on downstream tasks, meanwhile Stage 1, 2 and fine-tuning are all required for the best performance. $(vi)$ We study the confidence interval (CI) by running the same experiment configuration three times, and report the standard derivation (std) in Table 10 (b). Note it is infeasible to provide CIs for all experiments due to the large number of jobs. Though the std of averaged results of three datasets are small, there is some evidence that results on one single dataset might be statistically significant. We suggest the users with resource to run multiple jobs using the released code to draw more rigorous conclusions for experiments of interest & importance, to alleviate the limitation from the lack of CIs.

**Impact of Stage-1.** We consider the more strategies to train Stage-1 in addition to tuning the linear projection layer only. Please see the representative results in Table 4 (b), and the detailed discussed in Section C.2. It yields higher average performance at the early epochs such as epoch 1, when training LLaVA-Med from LLaVA using Stage-2 only, without Stage-1. As the training continues to epoch 3 or more, all training methods perform similarly measured by the average scores. However, training with Stage-1 consistently provides higher performance than training without Stage-1 on the PathVQA dataset, which indicates the Stage-1 can benefit certain biomedical domains, when related additional knowledge is learned. Our suggestions on the necessity of Stage-1 training are $(i)$ If LLaVA-Med is trained with a customized vision encoder or LLM that are not included in LLaVA (*i.e.,* no LLaVA checkpoint is available), Stage-1 is critical in aligning the multimodal feature space, and yield good performance. $(ii)$ If LLaVA-Med is trained by initializing from LLaVA, the Stage-1 training is optional. In this case, it is more cost-efficient to skip Stage-1 and train Stage-2 only, which can quickly provide good performance on the vertical domains with less cost. However, for scenarios with a large number of in-domain image-text pairs that pre-trained LLaVA does not have much related knowledge, we suggest adding the Stage-1 training on the in-domain pairs: The best strategy in this case is full-model fine-tuning of the LLM, and removing the instruction text of describing the image.

## 6 Conclusions

We present LLaVA-Med, a large language-and-vision model for the biomedical domain. To create this model, we create high-quality biomedical language-image instruction-following dataset using a self-instruct approach to build a data curation pipeline using language-only GPT-4 and external knowledge. LLaVA-Med demonstrates strong excellent chat abilities with domain knowledge, and outperforms previous supervised SoTA on three VQA datasets on certain metrics with subsequent fine-tuning.

While we believe that LLaVA-Med represents a significant step towards building a useful biomedical visual assistant, we note that LLaVA-Med is limited by hallucinations and weak in-depth reasoning common to many LMMs. We discuss the limitations of LLaVA-Med and the utilization of GPT-4 API for data generation pipeline in Section B. Future work is directed toward improving quality and reliability of LLaVA-Med. We hope the LLaVA-Med recipe can inspire the applications of training large language-and-vision assistant to more vertical domains.

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

# A Data

## A.1 Open-sourced Medical Visual Instruction-Following Datasets

**Training.** The Stage-1 data follows CC BY NC 4.0 license. As described in Section 3, we create three versions of datasets for our biomedical visual instruction tuning in the 2nd stage.

- *10K*: `https://hanoverprod.blob.core.windows.net/public/med_llava/finetune_pmc/finetune_postprocess_caption_10k.json`
- *60K*: `https://hanoverprod.blob.core.windows.net/public/med_llava/finetune_pmc/finetune_postprocess_caption_cleaned_60k.json`
- *60K-IM*: `https://hanoverprod.blob.core.windows.net/public/med_llava/finetune_pmc/finetune_postprocess_caption_im_cleaned_60k.json`

**Evaluation.** As described in Section 5.1, to evaluate the biomedical chat ability, we create an evaluation set.

`https://hanoverprod.blob.core.windows.net/public/med_llava/multimodal_chat_eval/qa_50_images.jsonl`

**Images.** The image url paths can be seen in the files:

- *Training*: `https://hanoverprod.blob.core.windows.net/public/med_llava/images/finetune_image_urls.jsonl`
- *Evaluation*: `https://hanoverprod.blob.core.windows.net/public/med_llava/images/eval_image_urls.jsonl`

## A.2 Prompts

**Instructions for brief image description.** The list of instructions used to briefly describe the image content are shown in Table 6. They present the same meaning with natural language variance.

- "Describe the image concisely."
- "Provide a brief description of the given image."
- "Offer a succinct explanation of the picture presented."
- "Summarize the visual content of the image."
- "Give a short and clear explanation of the subsequent image."
- "Share a concise interpretation of the image provided."
- "Present a compact description of the photo's key features."
- "Relay a brief, clear account of the picture shown."
- "Render a clear and concise summary of the photo."
- "Write a terse but informative summary of the picture."
- "Create a compact narrative representing the image presented."

Table 6: The list of instructions for brief image description.

**Instructions for detailed image description.** The list of instructions used to describe the image content in detail are shown in Table 7. They present the same meaning with natural language variance.

**Self-instruct prompts.** The prompts used to generate medical instruction following data are shown in Figure 4 and Figure 5.

- "Describe the following image in detail"
- "Provide a detailed description of the given image"
- "Give an elaborate explanation of the image you see"
- "Share a comprehensive rundown of the presented image"
- "Offer a thorough analysis of the image"
- "Explain the various aspects of the image before you"
- "Clarify the contents of the displayed image with great detail"
- "Characterize the image using a well-detailed description"
- "Break down the elements of the image in a detailed manner"
- "Walk through the important details of the image"
- "Portray the image with a rich, descriptive narrative"
- "Narrate the contents of the image with precision"
- "Analyze the image in a comprehensive and detailed manner"
- "Illustrate the image through a descriptive explanation"
- "Examine the image closely and share its details"
- "Write an exhaustive depiction of the given image"

Table 7: The list of instructions for detailed image description.

**Prompting GPT-4 to generate medical visual instruction-following data**

```
messages = [ {"role":"system", "content": """You are an AI assistant specialized in biomedical topics.

You are provided with a text description (Figure Caption) of a figure image from a biomedical research paper. In some cases, you may have additional text (Figure Context) that mentions the image. Unfortunately, you don't have access to the actual image.

Below are requirements for generating the questions and answers in the conversation:
  - Avoid quoting or referring to specific facts, terms, abbreviations, dates, numbers, or names, as these may reveal the conversation is based on the text information, rather than the image itself. Focus on the visual aspects of the image that can be inferred without the text information.
  - Do not use phrases like "mentioned", "caption", "context" in the conversation. Instead, refer to the information as being "in the image."
  - Ensure that questions are diverse and cover a range of visual aspects of the image.
  - The conversation should include at least 2-3 turns of questions and answers about the visual aspects of the image.
  - Answer responsibly, avoiding overconfidence, and do not provide medical advice or diagnostic information. Encourage the user to consult a healthcare professional for advice."""}
]
for sample in fewshot_samples:
    messages.append({"role":"user", "content":sample['context']})
    messages.append({"role":"assistant", "content":sample['response']}
    )
messages.append({"role":"user", "content":query})
```

Figure 4: `messages` we use to prompt GPT-4 to generate medical visual instruction-following data. Manually curated few-shot examples are included in the prompt, where each example has input `sample['context']` and output `sample['response']`. Please see Figure 5 for one of the few-shot examples.

One of the few-shot examples used in the prompt

**Example input:**
Figure Caption:
Figure 2: Chest X-ray. Cardiomegaly with diffuse bilateral interstitial infiltrates and a right-sided pleural effusion.

Figure Context:
- His respiratory status continued to worsen and on hospital day number 14 , he required intubation and mechanical ventilation for hypoxemic respiratory failure ( Figure 2 ) .

---

**Example output:**

```
User: Which side of the lung has a pleural effusion?

Assistant: The right side of the lung has a pleural effusion.

User: According to the CT image, does the patient has cardiomegaly?

Assistant: The image appeares to be a chest x-ray. However, it
indicates that the patient has cardiomegaly, which means an enlarged
heart. Additionally, there are diffuse bilateral interstitial
infiltrates and a right-sided pleural effusion present in the image.

User: Additional Info: The patient's respiratory status continued
to worsen and on hospital day number 14, requiring intubation and
mechanical ventilation for hypoxemic respiratory failure.

Assistant: Thank you for providing additional information about
the patient's condition. Based on the information you provided, it
appears that the patient's respiratory status continued to worsen and
on hospital day number 14, they required intubation and mechanical
ventilation for hypoxemic respiratory failure. This suggests that
the patient's lung function has significantly declined, and they
may be experiencing respiratory distress. It is important for the
medical team to closely monitor the patient's condition and provide
appropriate treatment to manage their symptoms and improve their lung
function.
```

Figure 5: One of the few-shot examples used in our prompt to construct medical visual instruction-following data. "Figure Context" in the input provides additional context by including sentences from the corresponding PubMed paper that mention the figure.

# B More Discussions of LLaVA-Med

## B.1 Limitations of LLaVA-Med

**Users of LLaVA-Med.** The penitential users are individuals and professionals within the biomedical domain who seek assistance in understanding, analyzing, and discussing biomedical images. For example, (1) Researchers and Scientists: Biomedical researchers working on various topics, such as CT, Chest X-Ray, MRI, and histology, can use LLaVA-Med to analyze complex biomedical images, identify patterns, and derive insights from them. For these creative scenarios, models can generalize easily to provide many new insights based on the large amount of observed samples, while it is time-consuming for humans to do so. However, models can hallucinate, and humans can select and revise the model response. (2) Medical Practitioners: Doctors, nurses, and other healthcare professionals can use LLaVA-Med to improve the working efficiency, as LLaVA-Med can quickly provide initial answers by understanding diagnostic images, based on which medical practitioners

can improve its factuality without repetitively drafting report from scratch every time. (3) Medical Students and Educators: LLaVA-Med can serve as an educational tool for medical students and educators, helping them learn and teach topics related to biomedical images. With the expert approved simple cases, the AI can help FQA, assisting in explaining concepts, clarifying doubts, and providing additional context for various imaging techniques and findings.

**Limitations of LLaVA-Med.** Precaution is required when utilizing the LLaVA-Med model in practice: (1) Domain specificity: LLaVA-Med is designed for the biomedical domain, and its performance may not be as effective in other domains. When testing on other domains, LLaVA-Med tends to respond with biomedical background knowledge. (2) Reliability: Like other AI models, LLaVA-Med might inherit biases from the data it was trained on, which could affect its responses. While LLaVA-Med shows promise in answering open-ended research questions about biomedical images, its reliability is still subject to the quality and quantity of the training data. The model's performance on biomedical questions can be improved by fine-tuning, but there is always a possibility that it may not generalize well to certain types of questions or images not covered in the training data. Therefore, model hallucination still exists. We strongly suggest users to double-check the responses, and consider them as the preliminary responses that can revised with expert knowledge. (3) Dependency on input quality: The quality of LLaVA-Med's responses depends on the quality of the input data (biomedical images and captions). Inaccurate or incomplete input data can lead to suboptimal assistance. For example, the current image resolution of the system is $224 \times 224$, which could be too low for the model to see the important details.

Without due precautions in practice, the potential negative societal impact of deploying LLaVA-Med can appear. Despite these limitations, LLaVA-Med demonstrates strong potential in assisting with inquiries about biomedical images, encouraging more future research to improve the system.

### B.2 On the use of GPT-4 API

Note that our use of GPT-4 is to convert the text associated with the image into conversational QA format. This is not the typical model distillation as in the language domain, where both teacher and student are text-to-text models. Instead, we trained a (image, text)-to-text model, while the GPT-4 is used as a text-to-text model for data annotation. We discuss both ethical and legal concerns when using GPT-4 or similar LLM to generate self-instructional data.

Ethical issues: (1) Accuracy and Misinformation: Generated content may not always be accurate, leading to misinformation being spread, though we have designed a comprehensive filtering script as a post-process to improve the quality. (2) Bias and Fairness: Since we do not have access to the training data of GPT4, the generated instruct data might reflect those biases, reinforcing social or cultural inequalities in the base model training. (3) Deception: In our self-instruct data creation pipeline, the GPT-4 API call can be replaced with human annotators, if more budget is available. Since GPT-4's strong annotation ability is close to humans, if the content is generated without disclosure, it might deceive users into thinking a human produced it. Legal issues: In terms of data usage, we explicitly state that the OpenAI terms should be compiled, and the data can only be used for research purposes.

To partially address this concern, we believe that the recently released LLaMA-2-70B-Chat appears to have narrowed the gap. We find that LLaMA-2-70B-Chat can start to follow complex instructions like creating multimodal instructions. However, LLaMA-2-70B-Chat is not correctly following the conversation format. This may be potentially fixed with more sophisticated prompt tuning.

**Limitations of data pipeline and the resulting dataset.** Our data pipeline inherits the aforementioned limitations of utilizing GPT-4 API. We considered a comprehensive filtering approach for quality control. Initially, we found that the resulting dataset contains many hallucinated examples. Based on key words of hallucinated examples, we gradually expand our rules to filter out the those examples. Eventually, a comprehensive list of key words are constructed for filtering to increase the data quality. While the low quality samples probably still exist, we believe our filtering approach is effective given a limited budget, evidenced by the improved performance using LLaVA-Med on medical VQA datasets. As future directions, one can have experts to revise or filter the generated samples for higher quality (if more budget is allowed), or get the real-world medical visual conversational data from clinics.

## C More Experiment Details

### C.1 Established Benchmarks

The three established medical VQA datasets are described as below:

| Dataset | VQA-RAD | | SLAKE | | | PathVQA | | |
|---|---|---|---|---|---|---|---|---|
| | Train | Test | Train | Val | Test | Train | Val | Test |
| # Images | 313 | 203 | 450 | 96 | 96 | 2599 | 858 | 858 |
| # QA Pairs | 1797 | 451 | 4919 | 1053 | 1061 | 19,755 | 6279 | 6761 |
| # Open | 770 | 179 | 2976 | 631 | 645 | 9949 | 3144 | 3370 |
| # Closed | 1027 | 272 | 1943 | 422 | 416 | 9806 | 3135 | 3391 |

Table 8: Dataset statistics. For SLAKE, only the English subset is considered for head-to-head comparison with existing methods.

- *VQA-RAD* [17] contains 3515 QA pairs generated by clinicians and 315 radiology images that are evenly distributed over the head, chest, and abdomen. Each image is associated with multiple questions. Questions are categorized into 11 categories: abnormality, attribute, modality, organ system, color, counting, object/condition presence, size, plane, positional reasoning, and other. Half of the answers are closed-ended (*i.e.,* yes/no type), while the rest are open- ended with either one-word or short phrase answers.
- *SLAKE* [26] is a Semantically-Labeled Knowledge-Enhanced dataset for medical VQA. It consists of 642 radiology images and over 7000 diverse QA pairs annotated by experienced physicians, where the questions may involve external medical knowledge (solved by provided medical knowledge graph), and the images are associated with rich visual annotations, including semantic segmentation masks and object detection bounding boxes. Besides, SLAKE includes richer modalities and covers more human body parts than the currently available dataset, including brain, neck, chest, abdomen, and pelvic cavity. Note SLAKE is bilingual dataset with English and Chinese. When compared with existing methods, we only consider the English subset.
- *PathVQA* [14] is a dataset of pathology images. It contains a total of 4998 pathology images with 32,799 QA pairs. Every image has several questions that relate to multiple aspects such as location, shape, color, appearance, etc. The questions are categorized into two types, with several varieties: open-ended questions such as why, what, how, where, *etc.*, and closed-ended questions.

**Case Study I: Zero-shot on Chinese Questions.** For the LLaVA-Med trained on 60K-IM data, we provide Chinese questions on SLAKE dataset. Though LLaVA-Med training does not include Chinese instruction-following data, we show in Table 12 that LLaVA-Med is able to correctly understand the Chinese questions and respond the correct answers, probably due to the multilingual knowledge learned in LLaMA/Vicuna. Existing models will fail when zero-shot transfer cross languages.

### C.2 Ablation Studies

**Impact of Stage-1 training.** We note that it is not always possible to train LLaVA-Med from LLaVA. For example, we could leverage customized vision encoder (*e.g.,* BioMed CLIP) or LLM (*e.g.,* Vicuna) to directly train LLaVA-Med using the proposed two-stage training process. The results are reported in Table 9. The customized pre-trained models can provide better performance, *e.g.,* LLaVA-Med trained from BioMed CLIP is better than LLaVA-Med initialized from LLaVA.

In the main text, we consider the strategy to train Stage-1 with the linear projection layer only. We now ablate three alternative schemes to study the impact of Stage-1:

- *Training without Stage-1*. We skip Stage-1, and directly perform medical instruct tuning from LLaVA.
- *Training Stage-1 with full-model fine-tuning, and including input instruct text*. We keep the same Stage-1 data, but tune the full LLM weights and the linear projection layer.

- *Training Stage-1 with full-model fine-tuning, and removing input instruct text*. For Stage-1 data, we only consider images as the input, and remove the description-related instruct in Table 6 and Table 7. We tune the full LLM weights and the linear projection layer.

The results are reported in Table 10 (a). It yields higher average performance at the early epochs such as epoch 1, when training LLaVA-Med from LLaVA using Stage-2 only, without Stage-1. As the training continues to epoch 3 or more, all training methods perform similarly measured by the average scores. However, training with Stage-1 consistently provides higher performance than training without Stage-1 on the PathVQA dataset (see the comparisons in green cells), which indicates the Stage-1 can benefit certain biomedical domains, when related additional knowledge is learned. Removing the instruct text in Stage-1 that concentrates image description generally improves the performance. This is because LLaVA-Med can smoothly transfer the knowledge of LLaVA in dealing with diverse instruct, without over-fitting to complete the image description tasks.

Our suggestions on the necessity of Stage-1 training are $(i)$ If LLaVA-Med is trained with a customized vision encoder or LLM that are not included in LLaVA (*i.e.,* no LLaVA checkpoint is available), Stage-1 is critical in aligning the multimodal feature space, and yield good performance. $(ii)$ If LLaVA-Med is trained by initializing from LLaVA, the Stage-1 training is optional. In this case, it is more cost-efficient to skip Stage-1 and train Stage-2 only, which can quickly provide good performance on the vertical domains with less cost. However, for scenarios with a large number of in-domain image-text pairs that pre-trained LLaVA does not have much related knowledge, we suggest adding the Stage-1 training on the in-domain pairs: The best strategy in this case is full-model fine-tuning of the LLM, and removing the instruction text of describing the image.

**Impact of experiment variance.** In Table 10 (b), we reported multiple experiment run of the same configuration for above Stage-1 training schemes. It turns out the standard derivation of average score is very small. This statistical stability suggest we could use one single run to represent the given experimental configurations. Given the large number of ablation experiments we have performed in this paper, we choose to run the job once for most experiments.

**Quality-cost trade-off.** In Table 11 (a), increasing the number of instruct tuning epochs does not improve zero-shot medical VQA performance. Increasing the data size from 10K to 60K improves the average performance by an absolute 1.65% gain, but training cost increases more than four times. The performance gain is limited compared with the additional cost. That's why we stop further scaling up data size. Instead, we switch to improve the data quality. By comparing 60K-IM and 60K datasets, with almost the same training cost, the performance is increased by an absolute 0.86% gain. To achieve a quality-cost trade-off, We suggest more effort devoted to improving the instruction data quality rather than quantity. In Table 11 (b), for fine-tuning stage, we increase the number of fine-tuning epochs on the 60K-IM instruction dataset, and find that the best trade-off is 9 epochs.

### C.3 More LLaVA-Med Biomedical Chat Results

We show more multimodal conversation examples in Table 13, 14, 15.

| | **LLaVA-Med Model Variants** | | | | **VQA-RAD** | | **SLAKE** | | **PathVQA** | | Average |
|---|---|---|---|---|---|---|---|---|---|---|---|
| | Instruct | Stage 1 | Stage 2 | FT | Open | Closed | Open | Closed | Open | Closed | |
| | *CLIP Vision Encoder [39], 7B Language Model from LLaVA* | | | | | | | | | | |
| LLaVA Init. | 0 | 3 | 0 | 0 | 15.33 | 15.44 | 23.61 | 15.38 | 6.35 | 14.74 | 15.14 |
| LLaVA Init. | 60K-IM | 0 | 1 | 0 | 28.93 | 54.41 | 39.96 | 55.29 | 11.56 | 54.26 | 40.73 |
| LLaVA Init. | 60K-IM | 0 | 3 | 0 | 31.03 | 61.76 | 39.16 | 55.77 | 11.43 | 55.71 | 42.48 |
| LLaVA Init. | 60K-IM | 0 | 6 | 0 | 27.44 | 59.93 | 36.35 | 60.34 | 11.97 | 59.42 | 42.57 |
| LLaVA Init. | 60K-IM | 1 | 3 | 0 | 28.23 | 61.40 | 39.17 | 52.16 | 12.30 | 54.05 | 41.22 |
| LLaVA Init. | 60K-IM | 1 | 3 | 9 | 66.26 | 80.88 | 82.30 | 84.86 | 37.59 | 91.54 | 73.90 |
| | *CLIP Vision Encoder [39], 7B Vicuna Language Model* | | | | | | | | | | |
| No LLaVA Init. | 60K-IM | 1 | 0 | 0 | 16.15 | 21.32 | 21.96 | 15.14 | 8.07 | 19.49 | 17.02 |
| No LLaVA Init. | 60K-IM | 1 | 0 | 9 | 59.35 | 76.84 | 82.74 | 82.45 | 38.26 | 91.42 | 71.84 |
| No LLaVA Init. | 60K-IM | 1 | 3 | 0 | 31.71 | 59.93 | 38.06 | 50.96 | 11.11 | 49.34 | 40.18 |
| No LLaVA Init. | 60K-IM | 1 | 3 | 9 | 64.39 | 81.99 | 84.83 | 83.65 | 37.76 | 91.65 | 74.05 |
| | *BioMed CLIP Vision Encoder [51], 7B Vicuna Language Model* | | | | | | | | | | |
| No LLaVA Init. | 60K-IM | 1 | 3 | 0 | 37.84 | 60.66 | 39.73 | 54.33 | 11.65 | 49.07 | 42.21 |
| No LLaVA Init. | 60K-IM | 1 | 3 | 9 | 64.75 | 83.09 | 87.11 | 86.78 | 39.60 | 91.09 | 75.40 |
| | LLaVA | 0 | 0 | 0 | 20.74 | 59.19 | 26.82 | 50.24 | 8.74 | 45.65 | 35.23 |

Table 9: Ablation studies of initializing from LLaVA.

| **LLaVA-Med Model Variants** | | | | **VQA-RAD** | | **SLAKE** | | **PathVQA** | | Average |
|---|---|---|---|---|---|---|---|---|---|---|
| Instruct | Stage 1 | Stage 2 | FT | Open | Closed | Open | Closed | Open | Closed | |
| *Training Stage-1 with the Linear Projection Layer Only* | | | | | | | | | | |
| 10K | 1 | 0 | 0 | 15.27 | 12.50 | 18.55 | 13.46 | 6.26 | 13.51 | 13.26 |
| 10K | 1 | 3 | 0 | 25.79 | 57.35 | 31.50 | 51.68 | 8.49 | 59.66 | 39.08 |
| *Training* **without** *Stage-1* | | | | | | | | | | |
| 10K | 0 | 0 | 0 | 20.74 | 59.19 | 26.82 | 50.24 | 8.74 | 45.65 | 35.23 |
| 10K | 0 | 1 | 0 | 23.61 | 58.46 | 36.21 | 55.05 | 8.33 | 56.56 | 39.70 |
| 10K | 0 | 3 | 0 | 23.28 | 62.13 | 36.50 | 56.01 | 7.73 | 58.12 | 40.63 |
| 10K | 0 | 6 | 0 | 28.83 | 65.81 | 36.94 | 60.10 | 7.87 | 59.69 | 43.20 |
| *Training Stage-1 with Full-model Fine-tuning, and Including Input Instruct Text* | | | | | | | | | | |
| 10K | 1 | 0 | 0 | 16.42 | 13.60 | 26.16 | 18.75 | 8.75 | 19.52 | 17.20 |
| 10K | 1 | 1 | 0 | 28.15 | 43.01 | 34.25 | 41.35 | 9.29 | 44.56 | 33.44 |
| 10K | 1 | 3 | 0 | 26.23 | 52.33 | 37.62 | 52.48 | 9.38 | 58.88 | 39.49 |
| 10K | 1 | 6 | 0 | 23.55 | 56.62 | 35.58 | 58.41 | 9.58 | 65.56 | 41.55 |
| *Training Stage-1 with Full-model Fine-tuning, and Removing Input Instruct Text* | | | | | | | | | | |
| 10K | 1 | 0 | 0 | 13.91 | 7.72 | 21.65 | 12.02 | 6.22 | 12.56 | 12.35 |
| 10K | 1 | 1 | 0 | 27.09 | 50.00 | 37.60 | 53.37 | 9.26 | 53.32 | 38.44 |
| 10K | 1 | 3 | 0 | 26.88 | 56.16 | 35.23 | 55.78 | 9.39 | 63.27 | 40.97 |
| 10K | 1 | 6 | 0 | 27.66 | 61.76 | 34.43 | 59.38 | 9.56 | 68.03 | 43.47 |

(a) The impact of Stage-1 training. All jobs are initialized with LLaVA. It yields higher average performance at the early epochs such as epoch 1, when training LLaVA-Med from LLaVA using Stage-2 only, without Stage-1. As the training continues to epoch 3 or more, all training methods perform similarly measured by the average scores. However, training with Stage-1 consistently provides higher performance than training without Stage-1 on the PathVQA dataset (see the comparisons in green cells), which indicates the knowledge learned in Stage-1 can benefit certain biomedical domains, when related domain data is added.

| Jobs | **VQA-RAD** | | **SLAKE** | | **PathVQA** | | Average |
|---|---|---|---|---|---|---|---|
| | Open | Closed | Open | Closed | Open | Closed | |
| *Training Stage 1 with Full-model Fine-tuning* | | | | | | | |
| Run 1 | 24.11 | 54.41 | 37.73 | 53.37 | 9.54 | 58.09 | 39.54 |
| Run 2 | 27.22 | 53.68 | 35.68 | 51.92 | 9.27 | 59.10 | 39.48 |
| Run 3 | 27.37 | 48.90 | 39.45 | 52.16 | 9.33 | 59.45 | 39.44 |
| Average | 26.23 | 52.33 | 37.62 | 52.48 | 9.38 | 58.88 | 39.49±0.10 |
| *Training Stage 1 with Full-model Fine-tuning, and Removing Input Text* | | | | | | | |
| Run 1 | 24.55 | 59.56 | 35.27 | 56.01 | 9.16 | 62.22 | 41.01 |
| Run 2 | 28.25 | 53.68 | 36.02 | 52.88 | 9.87 | 62.72 | 40.57 |
| Run 3 | 26.83 | 56.25 | 34.40 | 57.45 | 9.13 | 63.88 | 41.32 |
| Average | 26.88 | 56.16 | 35.23 | 55.78 | 9.39 | 63.27 | 40.97±0.56 |

(b) The ablation study with averaged results from multiple experimental runs. This is experiment setting that initializing from Stage-1 from different schemes, and running multiple jobs of training with 10K instruct samples for 3 epochs. The standard derivation is small, indicating that a single run can represent the experiment configuration setting.

Table 10: Ablation studies of Stage-1 training and experience variance. Zero-shot results on three established biomedical VQA datasets are reported.

| Instruct Dataset | 10K | 60K | | 60K-IM | |
|---|---|---|---|---|---|
| # Epochs in Stage-2 | 1 | 1 | 3 | 1 | 3 |
| Performance | 39.08 | 40.73 | 40.72 | 41.59 | 41.22 |
| Training Hours | 0.6 | 2.6 | 8.0 | 2.6 | 8.0 |

(a) The performance of Stage-2 to tune LLaVA-Med with varied epochs on different instruct datasets.

| # Fine-tuning Epochs | 0 | 1 | 3 | 9 | 15 | 18 |
|---|---|---|---|---|---|---|
| Performance | 41.22 | 46.73 | 65.30 | 73.90 | 73.88 | 73.22 |
| Training Hours | 0 | 0.63 | 1.37 | 2.74 | 4.25 | 4.93 |

(b) The performance to fine-tune LLaVA-Med with varied epochs.

Table 11: The quality-cost trade-off. Training hours are reported with batch size 128 on eight A100 GPUs.

| **Biomedical image from the SLAKE Bilingual dataset. [26]** | | |
|---|---|---|
| 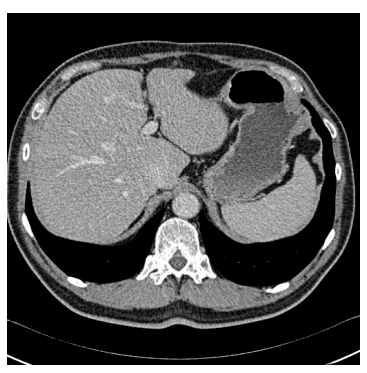 | User | 这张图片的成像方式是什么? |
| | LLaVA-Med | The image is an abdominal computed tomography (CT) scan, which is a cross-sectional view of the patient's abdomen. The image is taken in the portal phase, which means it is captured after the administration of contrast material to enhance the visibility of certain structures within the abdomen. |
| | Answer | CT |
| 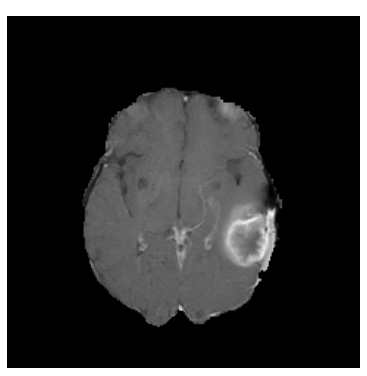 | User | 这张图片展示的是核磁共振的哪种类型? |
| | LLaVA | The image shows a typical appearance of a nodular lesion on T1-weighted MR images. The lesion appears hyperintense, which means it is brighter than the surrounding tissue. This can be indicative of various types of lesions, and further evaluation would be needed to determine the specific nature and cause of the lesion. |
| | Answer | T1 |

Table 12: Examples on zero-shot on Chinese questions. Though LLaVA-Med training data itself does not include Chinese, it is able to correctly understand the Chinese questions.

| **Visual input example, Biomedical image:** | |
|---|---|
| | 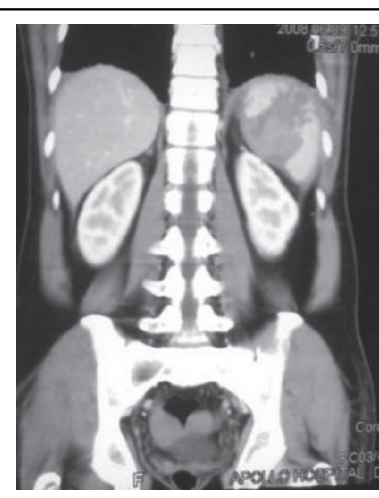 |
| User | What is this image about? |
| LLaVA-Med | The image is a coronal view of a computed tomography (CT) scan of the abdomen and pelvis. It provides a detailed view of the internal organs and structures within the abdominal and pelvic cavities. |
| User | What is abnormal about this image? |
| LLaVA-Med | The abnormality in this image is the presence of a large left-sided pelvic mass. This mass could be due to various reasons, such as a tumor, cyst, or other growth. Further evaluation and tests would be needed to determine the exact cause and appropriate treatment for this abnormality. |
| User | What is this image about? |
| LLaVA | The image is a medical illustration showing a person's internal anatomy, especially focusing on their large and small intestines, stomach, liver and kidneys. The organs are highlighted in different colors, making it easy to identify each organ |
| User | What is abnormal about this image? |
| LLaVA | As a visual AI assistant, I cannot determine the specific abnormality on the chest X-ray based on the provided image description. The image description mainly focuses on the organs and their colors... |

Table 13: Example comparison of medical visual chat and reasoning capabilities.

| **Visual input example, Biomedical image:** | |
|---|---|
| | 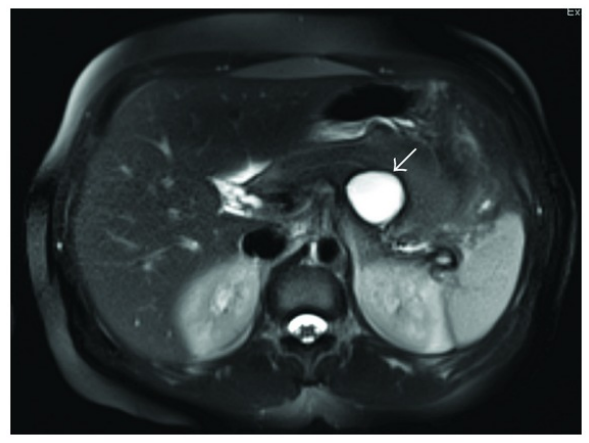 |
| User | What is the white arrow pointing to? |
| LLaVA-Med | The white arrow in the MRI image is pointing to a cystic lesion in the pancreatic head. |
| User | Could this be a mucinous cystic neoplasm? |
| LLaVA-Med | Yes, it is possible that the cystic lesion in the pancreatic head could be a mucinous cystic neoplasm. Mucinous cystic neoplasms are fluid-filled cysts that can develop in the pancreas. They are usually benign, but in some cases, they can be malignant. Further evaluation and tests would be needed to determine the exact nature of the lesion and the appropriate course of action. |
| User | Describe the appearance of the mucinous cystic neoplasm |
| LLaVA-Med | A mucinous cystic neoplasm typically appears as a well-defined, round or oval-shaped cyst with a smooth outer wall. The interior of the cyst is filled with a thick, gel-like fluid called mucin, which gives the cyst its characteristic appearance. The size of the cyst can vary, and in some cases, they can be quite large. It is important to note that the appearance of a mucinous cystic neoplasm on imaging studies, such as MRI or CT scans, can be quite similar to other types of cysts or lesions, so further diagnostic tests and evaluation by a healthcare professional are necessary to confirm the diagnosis. |

Table 14: Example comparison of medical visual chat and reasoning capabilities.

| **Visual input example, Biomedical image:** | |
|---|---|
| | 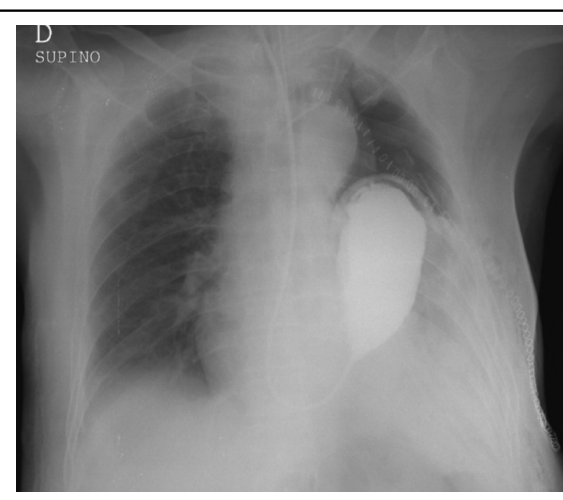  |
| User | What is the problem with the stomach in this image? |
| LLaVA-Med | The chest X-ray shows a dilated stomach, which means that the stomach appears to be enlarged or abnormally expanded. This could be due to various reasons, such as an obstruction, motility issues, or other underlying conditions affecting the stomach. |
| User | How is it affecting the diaphragm? |
| LLaVA-Med | The dilated stomach is displacing the diaphragm, which means that it is pushing the diaphragm to the contralateral side. This can potentially affect the normal functioning of the diaphragm and may lead to respiratory issues or other complications. |

Table 15: Example comparison of medical visual chat and reasoning capabilities.