# OpenReview forum: "LLaVA-Med: Training a Large Language-and-Vision Assistant for Biomedicine in One Day"
_NeurIPS.cc/2023/Track/Datasets_and_Benchmarks — NeurIPS 2023 Datasets and Benchmarks Spotlight_

### Official Review · Reviewer_73Lh · 2023-07-20
**Overall good work, but with some questions to answer first**

**Rating:** 5
**Confidence:** 4
**Clarity:** The paper is clear and easy-to-follow.

**Strengths:**

This paper makes a decent attempt in the underexplored direction of using multi-modal foundational models for biomedical applications. This line of research is highly relevant to the research community as researchers collectively explore the potential and limitations of foundational models.

The resources in this paper, including the dataset and the model checkpoints, could be useful for further biomedical applications to build on. Similarly, the data pipeline could inspire further works in other areas where large-scale instruction-tuning data is scarce.

The quality of research of this papr is good overall. The methodology of this paper is reasonably sound. A range of experiments are done on three different VQA datasets, including a ablation study that aims to probe the importance of different parts of the proposed dataset. Given the difficulty of evaluating open-ended text generation, the evaluation protocol in this paper is plausible.

**Additional Feedback:**

While I believe there are benefits that this paper could bring to the research community at large, I am leaning towards rejection, for there are some questions that cast doubts on some of the core results and therefore may undermine the main contributions.

**Correctness:**

Overall this paper has sound correctness. There are design choices of the dataset and the experiment evaluation that beg some questions, as detailed above.

**Documentation:**

No detail is provided regarding the availability, maintenance and responsible use of the dataset. The URLs provided in A.1 which supposedly link to the dataset are not accessible.

A public Github repository is available, however no code, trained model or dataset file is provided and therefore reproducibility cannot be ensured.

**Ethics:**

I do not believe there is ethical concern.

**Limitations:**

The authors adequately discussed the limitations of their model's performance and behavior.

I recommend the authors to further discuss the limitation and potential negative societal impact on two subjects. The first is the limitations of the methodology, in particular the data pipeline and the resulting dataset. The second is the potential negative societal impact of deploying LLaVA-Med, and the likes, in practice without due precautions.

**Opportunities For Improvement:**

### Methodology

#### GPT-4 generating training data

A crucial step of the data pipeline is using the text-only GPT-4 to generate simulated questions and answers based on the caption of an image (sometimes together with in-line mentions) but without the image itself. These synthesized conversations are then used to train the LLaVA-Med model.

Given the seemingly huge amount of knowledge GPT-4 stores in its parameters,  and its undesireble behavior of hallucination (e.g. see [1]), I suspect that, when generating answers, GPT-4 may include background knowledge that is not inferrable from the image and question provided. In fact, in the example provided in Figure 1, this is exactly the case, where in the answer to the last question "Can you explain more about the cavitating lesion?" only one sentence  contains information that can be inferred directly from the image ("In the context of the CT scan provided, the large cavitating lesion in the right upper lobe of the lung is suggestive of a malignant (cancerous) growth."). The rest of the answer seems to come from GPT-4's background knowledge.

It seems unfair to me to expect other models to provide the same amount and kind of background knowledge as GPT-4, when what is being evaluated is actually to answer from a provided image. Therefore, this casts some doubts on both the training process and the evaluation.

[1] Zhang, Muru, et al. "How language model hallucinations can snowball." arXiv preprint arXiv:2305.13534 (2023).

#### Training LLaVa-Med

Both visual encoder and the LM are frozen in stage 1, and only the visual encoder is frozen in stage 2. This design relies on the assumption that the encoders being frozen can generate good-quality representations for data, be it text or image, in the biomedical domain.

I have some doubt of this assumption, though I cannot state confidently that it is false. Still, I wonder if the performance can be improved with a different setup (e.g. train both encoders in stage 1), especially given that stage 1 training does more harm than good (details below).

### Experiments

#### Design

Important details of the experiments are not reported, e.g. hyper-parameters, random seeds and confidence intervals, etc.

#### Evaluation

When evaluating on general conversations (Section 5.1), GPT-4 is used both to provide gold answers and to score the model of interest (LLaVA and its variants).

While it is reasonable to expect that GPT-4's answers are overall better, given that the questions are generated from the captions, I have doubts on the fairness of using GPT-4 to score the answers. There can be various kinds of problems in using LLMs for evaluation [2-4], in particular self-enhancement, i.e. favoring its own generations.

[2] Li, Ruosen, Teerth Patel, and Xinya Du. "PRD: Peer Rank and Discussion Improve Large Language Model based Evaluations." arXiv preprint arXiv:2307.02762 (2023).

[3] Liu, Yang, et al. "Gpteval: Nlg evaluation using gpt-4 with better human alignment." arXiv preprint arXiv:2303.16634 (2023).

[4] Zheng, Lianmin, et al. "Judging LLM-as-a-judge with MT-Bench and Chatbot Arena." arXiv preprint arXiv:2306.05685 (2023).

#### The benefit of stage 1 training

In both sets of experiments (conversation and VQA), stage 1 seems to be detrimental to LLaVA-Med's performance, whichever metric. In particular, in Table 1, LLaVA-Med, if only trained in stage 1, performs worse than without both stage 1 and 2, overall as well as in all subgroups. Similarly, it performs substantially worse on all 3 VQA datasets with stage 1 training only (row 1-2 of Table 3b vs row 1 of Table 3a).

This further reinforces my doubts on whether freezing vision encoder and/or LM is optimal. The question of the benefit of stage 1 training is not discussed, and I look forward to the authors' response.

#### The benefit of stage 2 training

To a lesser extent, the benefit of stage 2 training is in doubt too. It is clear that doing so is beneficial, though it is unclear how much. The ablation study did not include the scenario where stage 2 training is dropped, and so no direct comparison is avaiable. However, it seems that the biggest increase in performance comes when fine-tuning is introduced or enhanced. This is the case for all 3 versions of the dataset (in Table 3b, line 5-6 vs line 3-4, line 9-10 vs line 7-8, line 13-17 vs line 11-12).

It would be very helpful to compare stage 2 training with directly fine-tuning.

**Relation To Prior Work:**

Overall, this paper has a decent discussion on how this work connects to and contrasts with previous works.

In addition to the lines of research already discussed, I recommend the authors to include prior works on instruction-tuning (e.g. dataset, methodology, etc.), as this is central to the contribution of this paper.

**Summary And Contributions:**

This paper proposes a data pipeline for vision-language instruction-tuning in the biomedical domain. Using this data pipeline, this paper creates a new dataset for training vision-language models, which are then transferred to downstream biomedical tasks such as medical Visual Question Answering (VQA).

The main contribution of this paper is threefold:

- a data pipeline that uses GPT-4 for generating instruction-tuning data without manual annotation

- a novel dataset for instruction-tuning vision-language models in the biomedical domain
- vision-language models and checkpoints obtained by training on such dataset and fine-tuning on downstream VQA datasets

---

> ### Author Response · Authors · 2023-08-25
> **Response to Reviewer 73Lh**
>
> > GPT-4 generating training data
>
> A: Please see our discussions of limitations of GPT-4 data generation pipeline in Q4 of the *General Response*. While we acknowledge the potential concerns, we emphasize that the significant performance improvement of  LLaVA-Med over LLaVA indicates the effectiveness of our data generation pipeline for biomedical domains.
>
> > Training LLaVa-Med
>
> A: Thanks for the suggestions, we ablate more strategies for Stage-1 training, and find that unfreezing the LLM weights in Stage-1 training, and further removing the text instruct in Stage-1 data is the most beneficial training strategy. Please see more discussions in Q2 in the *General Response*
>
> > Important details of the experiments are not reported, e.g. hyper-parameters, random seeds and confidence intervals, etc.
>
> We studied the standard derivation of the multiple jobs for the same experiment configuration in a new paragraph ``**Impact of experiment variance**'' (Lines 597-601) and Table 9(b) in Section C.2 of the updated Appendix. We found it is quite small. Therefore, we suggest using one single run to represent the experiment configuration due to the large number of experiments in our paper and the limited resources.
>
> The hyper-parameters for two stages are reported below, please see the codebase repo for more details.
>
> Hyperparameter | Global Batch Size | Learning rate | Epochs | Max length | Weight decay |
> | :--- | :----: | :----: | :----: | :----: | ---: |
> LLaVA-Med-7B (Stage-1) | 128 | 2e-3 | 1 | 2048 | 0 |
>
> Stage-2:
> Hyperparameter  | Global Batch Size | Learning rate | Epochs | Max length | Weight decay |
> | :--- | :----: | :----: | :----: | :----: | ---: |
> LLaVA-Med-7B (Stage-2) | 128 | 2e-5 | 3 | 2048 | 0 |
>
> >  Self-enhancement in GPT-4 Evaluation
>
> A: Thanks for bringing up the self-enhancement bias of LLMs for evaluation. While we agree with the existence of the bias, we believe that GPT-4 is a meaningful and consistent measurement in our settings. We use GPT-4 evaluation in our ablation study of LLaVA and LLaVA-Med variants trained from three instruct datasets, where GPT-4 is used in the data annotation process for all. Further, when computing relative scores in Table 1, we always compare the answers from the candidate model against the GPT-4’s answers. The ranking of the resulting numbers is consistent, though the numbers themselves might be biased to GPT’s answers. The conclusions of our ablation study can be obtained by considering the ranking. Further, by taking the self-enhancement bias into consideration for fairness, we expect that LLaVA-Med actually performs even closer to GPT-4 than the current numbers indicate.
>
> >The benefit of stage 1 training.
>
> A: Please see Q2 in the *General Response*. Overall, stage-1 training can be optional to save cost. However, there are cases that Stage-1 training is necessary for performance boost.
>
> >The benefit of stage 2 training
>
> A: We summarize the comparisons in the table below to ablate Stage-2. (i) In zero-shot settings, the two LLaVA-Med variants improve LLaVA by 5.40 and 7.25 points, respectively. (ii) In fine-tuning settings, the two LLaVA-Med variants improve LLaVA by 1.62 and 10.73 points, respectively. When the quality and quantality in the Stage-2 instruct is high, the gain of Stage-2 is larger.
>
> Model | Med Instruct Data | Stage-1 | Stage-2 | Fine-tuning | Average (Improvement)
> | :---| :--- | :----: | :----: | :----: | :---- |
> LLaVA | 0 | 0 | 0 | 0 | 35.23 |
> | | 0 | 0 | 0 | 3 | 54.57 |
> LLaVA-Med | 10K | 0 | 3 | 0 | 40.63  (+5.40)
> | | 10K | 1 | 3 | 0 | 39.08  (+3.85)
> | | 10K | 1 | 3 | 3 | 56.19  (+1.62)
> LLaVA-Med | 60K-IM  | 0 | 3 | 0 | 42.48 (+7.25)
> | | 60K-IM  | 1 | 3 | 0 | 41.22 (+5.99)
> | | 60K-IM  | 1 | 3 | 3 | 65.30 (+10.73)
>
> > I recommend the authors to further discuss the limitation and potential negative societal impact on two subjects. The first is the limitations of the methodology, in particular the data pipeline and the resulting dataset. The second is the potential negative societal impact of deploying LLaVA-Med, and the likes, in practice without due precautions.
>
> A: Please see Q3 and Q4 in the *General Response* to address the two limitations, respectively.

---

> > ### Comment · Reviewer_73Lh · 2023-08-25
> > **Thanks for the response**
> >
> > > Training LLaVa-Med
> >
> > Thanks for the additional experiments. I suggest also include the improved results in main text.
> >
> > > Important details of the experiments are not reported, e.g. hyper-parameters, random seeds and confidence intervals, etc.
> >
> > Thanks for the additional experiments. Indeed, variations across random seeds seem to be small enough.
> >
> > Still, the differences between runs might be statistically significant enough that one run's result is representative of the group, but the only way for readers to ascertain that is through authors providing confidence intervals. To me, this is still a (smaller) flaw that is not fully resolved.
> >
> > > Self-enhancement in GPT-4 Evaluation
> >
> > Thanks for acknowledging the bias of using GPT-4 for evaluation. It is indeed a good sign that the rankings are mostly preserved across settings. Please ensure to include this limitation in the main text.
> >
> > > The benefit of stage 1 training.
> >
> > Thanks for the additional ablation study. To me, the results indicate that indeed stage-1 training is not as valuable as suggested in the main text. I encourage authors to revise text in main text to reflect the limitations of stage-1 training (e.g. summarize the findings from Appendix C.2).
> >
> > > The benefit of stage 2 training
> >
> > Thanks for the additional experiments. These confirm that stage 2 is beneficial.
> >
> > However, my question was more about how important stage 2 is compared to fine-tuning. My doubt comes from the observation that fine-tuning seems to be the thing that provides the biggest improvement across the board (see original comment). Therefore, I suggested comparing stage 2 with direct fine-tuning (without stage 2). This would determine whether stage 2 training is necessary.
> >
> > To me, this is still a question that is not fully resolved.

---

> > > ### Author Response · Authors · 2023-08-26
> > > **Response to Reviewer 73Lh**
> > >
> > > Thanks for your quick feedback. We have revised the paper accordingly, and uploaded a new version. Due to the limited space, we can only select the most representative results and discussions to add in the main text.
> > >
> > > > Training LLaVa-Med
> > >
> > > The improved results are added in new Table 4(b) in the main text.
> > >
> > > > Important details of the experiments are not reported, e.g. hyper-parameters, random seeds and confidence intervals, etc.
> > >
> > > Thanks for your understanding. We emphasize that running confidence interval for all experiment configuration is not feasible given a large number of experiments in our paper, though it is more rigorous. The main points of our paper hold, with the small standard derivation we have shown for selected configurations.
> > >
> > > > Self-enhancement in GPT-4 Evaluation
> > >
> > > A new paragraph is added in Lines 218-225  in the main text.
> > >
> > > > The benefit of stage 1 training.
> > >
> > > A new paragraph is added in Lines 271-286  in the main text.
> > >
> > > > The benefit of stage 2 training.
> > >
> > > We now explicitly add that fine-tuning provides the biggest improvement. In the table we provided in the first response,  zero-shot of LLaVA yields 35.23, direct fine-tuning of LLaVA (without stage 2) yields 54.57, and adding Stage-2 yields 65.30. In Lines 267-270 of the main paper, we now add "When downstream samples are available, fine-tuning itself provides the largest performance gain (54.57-35.23=19.34). However, by training with high quality instruct data such as 60K-IM in the Stage-2, we can further boost performance significantly (65.30-54.57=10.73)"
> > >
> > > We would also like to point out that it is expected that fine-tuning with the full training set of a downstream dataset provides the most gain. The fact our instruct data significantly improves on top of fine-tuning is a strong evidence on the effectiveness of LLaVA-Med.

---

> > > > ### Comment · Reviewer_73Lh · 2023-08-28
> > > > **Some follow-up questions**
> > > >
> > > > Thanks for revising the manuscript.
> > > >
> > > > > Important details of the experiments are not reported, e.g. hyper-parameters, random seeds and confidence intervals, etc.
> > > >
> > > > I understand that it might be practically prohibitive to repeat experiments under every setting. However, I think my point still stands that confidence intervals are necessary for readers to gauge the statistical significance of the results.
> > > >
> > > > With that said, I think the best course of action now is to include the authors' argument (that it's infeasible to provide CIs for all experiments and there's some evidence to support that results might be statistically significant in Table 10b), while also acknowledging the limitation from the lack of CIs.
> > > >
> > > > > The benefit of stage 2 training.
> > > >
> > > > Upon reading the previous response again, I realize that there is indeed a direct comparison between stage 2 and fine-tuning (row 3 vs row 2, and row 6 vs row 2). In both cases, direct fine-tuning (row 2) performs better than stage 2 only (row 3 or 6).
> > > >
> > > > Indeed, as the authors pointed out, adding stage 1 and 2 (row 8) leads to a 10-point gain over fine-tuning (row 2). This showcases the benefit of the dataset. Taking all these into consideration, to me the fairer conclusion to draw on the value of stage 2 are: 1) stage 2 itself is not as effective as direct fine-tuning on downstream tasks, and 2) stage 1, 2 and fine-tuning are all required for the best performance.

---

> > > > > ### Author Response · Authors · 2023-08-28
> > > > > **Response to follow-up questions from Reviewer 73Lh**
> > > > >
> > > > > Thanks for your careful read and suggestions.
> > > > >
> > > > > > Important details of the experiments are not reported, e.g. hyper-parameters, random seeds and confidence intervals, etc.
> > > > >
> > > > > The suggested argument are added in the main text: Lines 270 - 276
> > > > >
> > > > > > The benefit of stage 2 training.
> > > > >
> > > > > The suggested conclusions are added in the main text: Lines 269 - 270
> > > > >
> > > > >
> > > > > Please let us know if you have any further suggestions to improve the paper.

---

### Official Review · Reviewer_kmHu · 2023-07-21
**Significant advancements have been made in multimodal research specialized for the biomedical domain.**

**Rating:** 7
**Confidence:** 3
**Clarity:** This paper is written very clearly, c…

**Strengths:**

1. They have enabled the learning of a more scalable multimodal answering model by adding a pipeline that can learn answers to open-ended questions from previously costructed  image-text pair data.
2. While this research was conducted in the biomedical domain, if it's possible to construct an image-caption dataset in various specialty fields, model adaptation is possible through the same pipeline.

**Additional Feedback:**

There is nothing more to add.

**Correctness:**

As the PMC dataset itself is open-source for research, there's a high probability that there's no problem with the derived datasets. The performance evaluation method appears to be appropriate.

**Documentation:**

Dataset and code cannot be found in the repository.

**Ethics:**

They used PMC based dataset which is opened for research.

**Limitations:**

Reproducibility is challenging as the dataset and code cannot be found in the repository.

**Opportunities For Improvement:**

The authors noted that the training time was short, which they described as important for reasonable training costs. However, performance improvements are expected through actual data or model scale-up (Table 3). Therefore, if there were experimental results at a data size of over 60K and an increased number of epochs, it would have been greatly helpful in anticipating the appropriate quality-cost trade-off.

**Relation To Prior Work:**

Yes it is. There is a model with a very similar concept, called Visual Med-Alpaca. They have explained this in detail, and also thoroughly described the essential differences.

**Summary And Contributions:**


There have been substantial academic and commercial advancements using general large image-text datasets. However, performance has been subpar or plagued by serious hallucinations in domains requiring specialized knowledge, including the biomedical field. To address this, the authors brilliantly constructed a dataset called PMC-15M from previous research. They developed a data generation pipeline that could create various sets (image, instruction, output) using GPT-4 in this work. Additionally, they constructed a model for open-ended questions through fine-tuning and additional training with LLaVa.
Through their experimental results, the authors demonstrated that the model developed through this research process had learned a higher level of answering capability than general language models. If these research findings are released as open-source, they could greatly contribute to multimodal model learning in the biomedical domain in the future.

---

> ### Author Response · Authors · 2023-08-25
> **Response to Reviewer kmHu**
>
> > The authors noted that the training time was short, which they described as important for reasonable training costs. However, performance improvements are expected through actual data or model scale-up (Table 3). Therefore, if there were experimental results at a data size of over 60K and an increased number of epochs, it would have been greatly helpful in anticipating the appropriate quality-cost trade-off.
>
> A: To address this question, we add a new paragraph ``**Quality-cost trade-off**'' (Lines 602-610) and Table 10 in Section C.2 of the updated Appendix.
>
> > Reproducibility is challenging as the dataset and code cannot be found in the repository.
>
> A: Please see Q1 in the *General Response*.

---

### Official Review · Reviewer_LUar · 2023-07-21
**One more step towards a language-vision open-source assistant**

**Rating:** 7
**Confidence:** 4
**Clarity:** In overall, the paper is well written…

**Strengths:**

- The paper presents a large ablation study in the Table 3.

- Can be easily reproducible thanks to the code on GitHub.

- Rapidly adaptable to newer LM and vision encoder thanks to its interchangeable components.


**Additional Feedback:**

No

**Correctness:**

The training datasets are constructed using the well-establish self-instruct method.

The selected tasks and models used for the evaluation sound good and are compared using relevant metrics.


**Documentation:**

The dataset collection is well detailed and the authors will make the project available on GitHub under the Apache 2 and CC BY NC 4.0 license (not available yet).

**Limitations:**

- The terms of use of OpenAI API are not respected "(c) Restrictions. You may not [...] (iii) use output from the Services to develop models that compete with OpenAI". (https://openai.com/policies/terms-of-use)

- Do you take only PMC OA data ? Otherwise, the models resulting from these data aren't meant for commercial use. You mention the CC BY NC 4.0 license on GitHub, but not on the paper itself.

**Opportunities For Improvement:**

- We don't have any information about the kind of errors made by the model or even how human are perceiving the GPT-4 generated instructions ? It would be interesting to do it on at least a subsample of a hundred of elements.

- Authors compared their method against a "stage 1" only training and observed better overall performances. This approach seem to be a more appropriated model adaptation method than only adapting it to the end domain. However, it would be interesting to this a comparison with the LLaVa model only trained on the "stage 2" to observe if the combination of LLaVa and instructions doesn't obtain similar performances with and without the "stage 1".

**Relation To Prior Work:**

The contribution is well contextualized and none of the important related works seem to be missing.

**Summary And Contributions:**

The paper present:

- A new dataset of multi-modal instructions on biomedical data obtained using self-instruct and GPT-4.

- A set of resulting open-sourced models.

- Authors set up an experimental protocol allowing to identify and isolate individual variables (data size, model size, ...).

- Evaluate the models on a set of multimodal tasks with different difficulties by variating closed and open QA against a large panel of SOTA architectures.

- Introducing a curriculum learning method to fine-tune the LLaVa model.

---

> ### Author Response · Authors · 2023-08-25
> **Response to Reviewer LUar**
>
> > We don't have any information about the kind of errors made by the model or even how human are perceiving the GPT-4 generated instructions ? It would be interesting to do it on at least a subsample of a hundred of elements.
>
> A: A type of error from LLaVA-Med is it may use its domain knowledge from the biomedical field, which is largely learned from the instruction data generated by GPT-4, to answer questions without fully utilizing the input image. Since during instruction generation, GPT-4 does not have access to the image, it may produce questions that are not answerable for the given image, and/or provide multiple plausible answers based on its biomedical domain knowledge, some of which may be coincidentally correct. This cause LLaVA-Med to sometimes rely more on its domain knowledge than on the image to answer the questions. To mitigate this problem, we intend to improve our prompt for instruction generation to ensure that GPT-4 generates instructions that are strictly grounded on the provided caption, so that the questions and answers are directly related to the caption and do not make guesses using external information. However, we also note that there may be cases where the caption itself contains information that is not inferable from the image alone. These noisy instructions lead the model to be less grounded on the image and more likely to hallucinate.
>
> The model has higher performance for image modalities and organ combinations that are highly represented in the PMC-OA dataset, such as chest X-ray and brain MRI, than for those that are underrepresented or rare. Moreover, the model fails to provide correct diagnosis when the image features are subtle and/or require reasoning. The model may also mistake the diagnosis condition for another condition affecting the same organ.
>
>
> > Authors compared their method against a "stage 1" only training and observed better overall performances. This approach seem to be a more appropriated model adaptation method than only adapting it to the end domain. However, it would be interesting to this a comparison with the LLaVa model only trained on the "stage 2" to observe if the combination of LLaVa and instructions doesn't obtain similar performances with and without the "stage 1".
>
> A: Please see Q2 in *General Response* for detailed discussion.  More specifically, please find the requested experiment comparisons in the newly added tables in the updated Appendix: Table 8 and Table 9(a).
>
> > The terms of use of OpenAI API are not respected "(c) Restrictions. You may not [...] (iii) use output from the Services to develop models that compete with OpenAI". (https://openai.com/policies/terms-of-use)
>
> A: Please see Q4 in *General Response* for detailed discussion. In terms of data usage, we explicitly state that the OpenAI terms should be compiled, and the data can only be used for research purposes, not for commercially competing purposes.
>
> > Do you take only PMC OA data ? Otherwise, the models resulting from these data aren't meant for commercial use. You mention the CC BY NC 4.0 license on GitHub, but not on the paper itself.
>
> A: Yes, we only use PMC-OA data with terms of use that are “non-commercial use only” or “commercial use allowed”. CC BY NC 4.0 license is added in the paper revision. Thanks.

---

### Official Review · Reviewer_1nwT · 2023-07-21
**Comments to the paper**

**Rating:** 9
**Confidence:** 4
**Clarity:** Yes

**Strengths:**


1 This work presents a novel data generation pipeline to create diverse (image, instruction, output) instances.
2 This work presents a new vision-language model which may benefit the related research community.
3 Both dataset and codes are open source.

**Additional Feedback:**

None.

**Correctness:**

The dataset was constructed in a sound way.
The evaluation methods and experiment design are appropriate and performed correctly.


**Documentation:**

Done.

**Ethics:**

None.

**Limitations:**

1 I doubt whether this model is useful for us and what kind of people needs this model.

**Opportunities For Improvement:**


1 Explain the three columns in Tables 3(a) and 3(b).
2 Give a deep analysis about the experimental result.

**Relation To Prior Work:**

Clearly discussed.

**Summary And Contributions:**

This paper proposes a cost-efficient approach for training a vision-language conversational assistant that can answer open-ended research questions of biomedical images, by leveraging a large-scale, broad-coverage biomedical figure-caption dataset extracted from PubMed Central, employing GPT-4 to 10 self-instruct open-ended instruction-following data from the captions, and fine-tuning a large general-domain vision-language model using a novel curriculum learning method.

---

> ### Author Response · Authors · 2023-08-25
> **Response to Reviewer 1nwT**
>
> > Explain the three columns in Tables 3(a) and 3(b).
>
> A: In Table 3(a), for each dataset, (1) “Ref”: the reference performance numbers of existing methods quoted from in the previous papers. (2) “Open” and “Closed”: the performance of our experiment job runs on the open-set and closed-set evaluation questions, respectively.
>
> In Table 3(b), the meaning of  “Open” and “Closed” for each dataset remain the same. Further, “Instruct” indicates which one of three instruct data sets is used,  “Stage 1”, “Stage 2” and “FT” indicate the number of epochs used to train LLaVA-Med in the Stage 1, Stage 2, and Downstream Fine-tuning Stage, respectively (see the Figure 3 for the visual illustration of three stages in different colored boxes).
>
> > Give a deep analysis about the experimental result.
>
> A: A new subsection C.2 (Lines 564-610) in the updated Appendix is added for more ablation studies and detailed experimental results, along with the deep analysis.
>
> > I doubt whether this model is useful for us and what kind of people needs this model.
>
> A: A new paragraph ``**Users of LLaVA-Med**'' (Lines 476-490) is added in subsection B.1 in the updated Appendix.

---

### Official Review · Reviewer_SDgf · 2023-07-23
**Really nice paper**

**Rating:** 8
**Confidence:** 4

**Strengths:**

- Great experiments spanning 3 datasets, multiple baseline LLMs of different param scales.
- Generic, data-centric method focusing on training set creation to adapt (in theory) any causal LLM for use in biomedical or other specialized domaines
- Paper is clearly written

**Additional Feedback:**

- N/A

**Clarity:**

- The paper is very clearly written

**Correctness:**

- The claims appear correct (or at least no gross errors are evident)

**Documentation:**

- Documentation is sufficient

**Ethics:**

- The authors should discuss some of the current debate around legality of using GPT-4 for self-instruction and the limitations that implies for potentially using this method in other domains, settings.

**Limitations:**

-  The most evidence limitation is the hard reliance on a GPT-4 caliber teacher model. There are ethical and legal questions still undecided on the ability to use GPT-4 to generate self-instruct training data.
- One potential negative impact would be further pushing the research community to rely on opaque, proprietary models as a cheap way to bootstrap special domain models. In fact what we disparately need in the research community to focus on stronger shared base models, inclusive of methods for training strong special domain models.

**Opportunities For Improvement:**

- I wasn't exactly clear on how closed-set questions were evaluated (section 5.2). Did you apply any constraints here (e.g. rank eval over some set of completions) or is it treated the same as open-set, but measured with accuracy? Lines 224-227 seem more like they are describing the closed set experiment protocol.

- Section 1(c) of the checklist "Did you discuss any potential negative societal impacts of your work?" and the statement "we discussed the safety and ethical filtering policy of our data generation process in Section 3" doesn't seem to match what is discussed in section 3. Could you elaborate on this connection -- there doesn't seem to be much text discussing this.

- It would be nice to characterize the performance of models beyond GPT-4 as teachers, given OpenAI's lack of transparency on it's training procedures.

- There is some recent work that questions some of the process of using stronger/larger models to generate training data for smaller models https://arxiv.org/abs/2305.15717  It would be nice if the authors could speak to this specific line of criticism

- None of the code or data is currently shared https://github.com/microsoft/LLaVA-Med/issues/5 I hope the authors will provide this code before the rebuttal period is complete so that reviewers and the community may test it.


**Relation To Prior Work:**

- Related work is comprehensive (for some a new area)
- Nice comparison to current SOTA (Table 3a)


**Summary And Contributions:**

The paper presents LLaVA-Med, a flexible multimodal (image/text) LLM adaptation process using paired image/caption data from PubMed to generate a self-instruct dataset via GPT-4. Their approach approach creates instruction data to align LLMs with biomedical concepts by sampling 600k image/caption pairs from PMC-15M and using prompting methods to generate conversational-like exchanges via GPT-4. Training their LLM then consists of a 2 stage process: (1) learn a projection matrix given a frozen LLM and vision encoder using an instruction task of predicting image captions (using image/caption pairs from PMC-15M and a simple instruction template) and (2) using multi-round conversation instruction data generated by GPT-4 and image captions to fine tune the LLM and projection matrix (keeping the vision encoder fixed). The net result is an LLM that is better aligned with medical instruction tuning use cases and in several cases matches SOTA or sets a new best score.

The paper looks at 3 visual QA datasets (VQA-RAD, SLAKE, PathVQA), several of LLMs and has several ablation studies based. LLaVA-Med clearly outperforms the base LLaVA after the 2nd round of instruction tuning. The overall training protocol benefits from more instruction training data (as to be expected).

---

> ### Author Response · Authors · 2023-08-25
> **Response to Reviewer SDgf**
>
> > I wasn't exactly clear on how closed-set questions were evaluated (section 5.2). Did you apply any constraints here (e.g. rank eval over some set of completions) or is it treated the same as open-set, but measured with accuracy? Lines 224-227 seem more like they are describing the closed set experiment protocol.
>
> A: For close-set, we evaluate the accuracy/percentage of the ground-truth tokens that appear in the generated sequences, which is quite similar to our open-set evaluation protocol. Lines 224-227 discusses how the open-set problem is reformulated as a closed set problem and evaluated by the closed set experiment protocol. We have clarified it in the revision (Lines 225-231).
>
> >Section 1(c) of the checklist "Did you discuss any potential negative societal impacts of your work?" and the statement "we discussed the safety and ethical filtering policy of our data generation process in Section 3" doesn't seem to match what is discussed in section 3. Could you elaborate on this connection -- there doesn't seem to be much text discussing this.
>
> This is briefly discussed in the end of ``Section 6: Conclusions''. We have added new subsections in Appendix to discuss the potential negative societal impacts of our work. Please see Q3 and Q4 in the *General Response*.
>
> > characterize the performance of models beyond GPT-4 as teachers, given OpenAI's lack of transparency on it's training procedures.
>
> A: In our experiments, we find that the capability of the teachers is crucial to the quality of the generated instruction-following data. For example, among proprietary models, we found GPT-4 shows stronger spatial understanding & reasoning ability than ChatGPT, and thus generates higher quality data. Therefore, we finally chose to use GPT-4 as the teacher in our data generation.
> The reviewer raises a good point about OpenAI's lack of transparency on its training procedures. More exploration is needed to consider open-source LLM as the teacher.
>
> To address this concern, we believe that the recently released LLaMA-2-70B-Chat appears to have narrowed the gap. We conducted a preliminary study on around 200 samples. Specifically, we generate 200 conversation samples for each category, using LLaMA-2-70B-Chat, ChatGPT, and GPT-4. We find that LLaMA-2-70B-Chat can start to follow complex instructions like creating multimodal instructions. However, LLaMA-2-70B-Chat is not correctly following the conversation format. This may be potentially fixed with more sophisticated prompt tuning. We then quantitatively evaluate the generated instructions using GPT-4 as the judge: (1) the correctness of the answers generated, and (2) the complexity of the instructions generated for complex reasoning questions.
>
> | | Correctness | Complexity |
> | :--- | :----: | :----: |
> | LLaMA-2-70B-Chat | 8.7 | 7.4
> | ChatGPT | 9.5 | 9.2
>
> Though the best open-source LLM still lags behind, these initial results are promising, and suggest that our recipe can be potentially applied to open-source LLM as the teacher, especially when their capabilities are improved in the future.
>
> > Criticism using stronger/larger models to generate training data for smaller models.
>
> A: Thanks for bringing it up. We are happy to discuss the false promise that the open LLMs could catch up with the proprietary LLMs raised in the paper. To align the discussions, we argue that there are two distinctive abilities for LLMs: the instruction-following ability to understand which task to perform, and massive knowledge storage to complete the task with quality. Imitation models are good at the former, by mimicking ChatGPT’s style but not its factuality. They authors conclude that there exists a substantial capability gap between open and closed LMs that, with current methods, can only be bridged using an unwieldy amount of imitation data or by using more capable base LMs. They also advocate that the highest leverage action for improving open-source models is to tackle the difficult challenge of developing better base LMs. We further note that the issue of lacking knowledge is particularly severe in the vertical domains such as healthcare, even for some commercial closed LLMs, developing base LMs for such vertical domains are valuable as well.
>
> However, unfortunately the resources to train such base LMs are only available in a few industry labs, and the formulas to train the base LMs is largely well explored. We kindly suggest that it seems more promising for most academic research labs to explore the opportunities in data creation, alignment research with affordable resources, or explore the techniques to reduce the compute barriers.
>
>
> >None of the code or data is currently shared
>
> A: Please see Q1 in the *General Response*.
>
> > Two limitations
>
> A:  Hope your concerns can be alleviated with our new discussion on open-source teacher models like LLaMA-2-70B and Q4 in the *General Response*.

---

> > ### Comment · Reviewer_SDgf · 2023-08-29
> >
> > Thanks to to the authors for running additional experiments using an alternative (LLaMA-2-70B-Chat) model to GPT-4 and releasing code and dataset preview. They authors have addressed my original critiques and I am still in strong favor of acceptance for this paper.

---

### Author Response · Authors · 2023-08-25
**General Response**

We sincerely thank all the reviewers for their time and their thoughtful comments and questions. We are encouraged that the reviewers find that:

-  The data-centric method and the proposed data generation pipeline is generic, novel and beneficial for the biomedical domain (1nwT, 73Lh), and can be generalized to various specialty fields (kmHu). The proposed LLaVA-Med model is modularized with interchangeable components (LUar).
- The paper is well written (SDgf). Our work is comprehensive in terms of experiments (SDgf,  LUar). The open source makes research reproducible.

We attempted our best to address the questions as time allowed. We believe the comments & revisions have made the paper stronger and thank all the reviewers for their help. We first address the shared questions in the general response. Please find individual responses to your questions below. More response with quantitive results will be provided once available.

***

**Q1**: Open source.

A: As authors of LLaVA-Med, we are eager to open source every aspect of the project immediately, so that the community can build upon. However, the LLaVA-Med project has to comply with the Microsoft release process for responsible AI. This process usually takes a long time due to the recent emergence of generative AI. Based on the conditional approval, we are now allowed to provide a link of the LLaVA-Med project repo for a selected group (reviewers and AC) at this moment and will be sharing in a separate response. We are confident the full project can be released very soon.

**Q2**: Impact of Stage-1 Training.

A: Thanks for bringing up this question. We comprehensively add more experiments in two more tables and describe our finding in the paragraph "**Impact of Stage-1 training**" (Line 565-596) in the updated Appendix Section C.2. Our suggestions on the necessity of Stage-1 training are summarized:
- (i) If LLaVA-Med is trained with a customized vision encoder or LLM that are not included in LLaVA (i.e., no LLaVA checkpoint is available), Stage-1 is critical in aligning the multimodal feature space, and yield good performance.
- (ii) If LLaVA-Med is trained by initializing from LLaVA, the Stage-1 training is optional. In this case, it is more cost-efficient to skip Stage-1 and train Stage-2 only, which can quickly provide good performance on the vertical domains with less cost. However, for scenarios with a large number of in-domain image-text pairs that pre-trained LLaVA does not have much related knowledge, we suggest adding the Stage-1 training on the in-domain pairs: The best strategy in this case is full-model fine-tuning of the LLM and removing the instruction text of describing images. The text and figure in the main paper is also revised accordingly.

**Q3**: Limitations of LLaVA-Med

A: The limitations are detailed in the updated Appendix Section B.1 "**Limitations of LLaVA-Med**" (Line 475-508)

**Q4**: Limitations of utilizing GPT-4 for Data Generation

A:   The limitations are detailed in the updated Appendix Section B.2 "**One the use of GPT-4 API**" (Line 509-537)

---

> ### Author Response · Authors · 2023-08-25
> **Open-Source Link**
>
> To compile with Microsoft release policy, we provide the open-source link for a selected group for the review purposes, and hope reviewers/AC do not distribute the link to a broader audience for the time being. In the link below, the repo contains all aspects of the LLaVA-Med project, including codebase, data, checkpoint and demo. We will release it to the public as we get the final approval in the near future.
>
> https://github.com/LLaVA-VL/LLaVA-Med-preview

---

### Comment · Area_Chair_6hw1 · 2023-08-29
**Reviewers, please check the rebuttal and update your scores**

Dear Reviewers,

The author-reviewer discussion period will end on Aug. 30th. Please check if the authors' rebuttal has addressed your concerns. If not, you can ask the authors for further clarification. Finally, please remember to update your scores if necessary.

Sincerely,

Area Chair

---

### Decision · Program_Chairs · 2023-09-22

**Decision:**

Accept (Spotlight)

**Comment:**

This manuscript received five reviews: four high scores (7 or higher) with one marginally negative score. The rebuttal addressed most concerns raised in the initial reviews. The manuscript was well written. The pros and cons (after rebuttal) identified by the reviewers are as follows.

Pros:
1.	This paper makes a decent attempt in the underexplored direction of using multi-modal foundational models for biomedical applications. This line of research is highly relevant to the research community as researchers collectively explore the potential and limitations of foundational models.
2.	This work presents a novel data generation pipeline to create diverse (image, instruction, output) instances.
3.	Comparison experiments are done on three different Med-VQA datasets, including an ablation study that aims to probe the importance of different parts of the proposed dataset.

Cons:
1.	The ground truth was collected by interacting with GPT-4, an opaque and proprietary model, which forbids a commercial application of the derived model. Nevertheless, the research on medical LLM is in its early stage, not ready for commercial application before finishing a thorough clinical evaluation.
2.	The last reviewer still has concerns about the impact of different training stages. However, this is a minor issue.